# Innovative Approaches in Cancer Treatment: Emphasizing the Role of Nanomaterials in Tyrosine Kinase Inhibition

**DOI:** 10.3390/pharmaceutics17060783

**Published:** 2025-06-16

**Authors:** Antónia Kurillová, Libor Kvítek, Aleš Panáček

**Affiliations:** Department of Physical Chemistry, Faculty of Science, Palacký University in Olomouc, 17. Listopadu 12, 77900 Olomouc, Czech Republic; antonia.kurillova01@upol.cz (A.K.); libor.kvitek@upol.cz (L.K.)

**Keywords:** nanomaterials, metal nanoparticles, gold nanoparticles, nanomedicine, tyrosine kinase inhibitors, drug delivery, anticancer treatment

## Abstract

Medical research is at the forefront of addressing pressing global challenges, including preventing and treating cardiovascular, autoimmune, and oncological diseases, neurodegenerative disorders, and the growing resistance of pathogens to antibiotics. Understanding the molecular mechanisms underlying these diseases, using advanced medical approaches and cutting-edge technologies, structure-based drug design, and personalized medicine, is critical for developing effective therapies, specifically anticancer treatments. **Background/Objectives**: One of the key drivers of cancer at the cellular level is the abnormal activity of protein enzymes, specifically serine, threonine, or tyrosine residues, through a process known as phosphorylation. While tyrosine kinase-mediated phosphorylation constitutes a minor fraction of total cellular phosphorylation, its dysregulation is critically linked to carcinogenesis and tumor progression. **Methods**: Small-molecule inhibitors, such as imatinib or erlotinib, are designed to halt this process, restoring cellular equilibrium and offering targeted therapeutic approaches. However, challenges persist, including frequent drug resistance and severe side effects associated with these therapies. Nanomedicine offers a transformative potential to overcome these limitations. **Results**: By leveraging the unique properties of nanomaterials, it is possible to achieve precise drug delivery, enhance accumulation at target sites, and improve therapeutic efficacy. Examples include nanoparticle-based delivery systems for TKIs and the combination of nanomaterials with photothermal or photodynamic therapies to enhance treatment effectiveness. Combining nanomedicine with traditional treatments holds promise and perspective for synergistic and more effective cancer management. **Conclusions**: This review delves into recent advances in understanding tyrosine kinase activity, the mechanisms of their inhibition, and the innovative integration of nanomedicine to revolutionize cancer treatment strategies.

## 1. Introduction

Tyrosine kinases (TKs) are generally protein enzymes belonging to the class of transferases, more specifically phosphotransferases, whose role is to transfer the phosphoryl group from ATP or other nucleoside phosphates to the amino acid tyrosine in proteins. This functional group transfer process is referred to as phosphorylation and is a reversible action. This transfer of a phosphoryl group to one or more amino acid residues leads to conformational changes and affects several physiological and biochemical processes. Protein tyrosine kinases are part of multicellular organisms and are enzymes that activate signaling pathways of cell proliferation. Protein tyrosine kinase signaling is an indispensable component of cellular communication in multicellular organisms. Human tyrosine kinase fuses signaling proteins controlling many biological processes ranging from cell growth, cell proliferation, cell survival, metabolism, migration, response to pathogens and to almost all stages of embryonic development. Aberrant signaling is associated with human diseases such as diabetes, bone disease, inflammatory diseases, and is a frequent concomitant of cancer [1,2]. Protein kinases catalyze a reaction in which the phosphoryl ion $\left({PO}_{3}^{2-}\right)$ is transferred from adenosine triphosphate (ATP) to a protein substrate:$${MgATP}^{1-}+protein-O:H\to protein-{O:PO}_{3}^{2-}+MgADP+H^{+}$$

Based on the phosphorylated OH group on the protein, we classify these catalysts as serine/threonine or tyrosine kinases. Most of the protein phosphorylation occurs at serine or threonine residues, with tyrosine phosphorylation accounting for less than 1% of total cellular phosphorylation [3,4]. The localization and duration of kinase activity is tightly regulated. Protein kinases are allosterically regulated enzymes, meaning that a perturbation at the allosteric site of the protein is transferred to the active, orthosteric site. Disruption of the allosteric site can be due to protein dimerization, post-translational modification, or ligand binding [5,6].

Depending on their position in the cell, tyrosine kinases can either be membrane-bound, called receptor tyrosine kinases (RTKs), or they can be free in the cytoplasm, called non-receptor tyrosine kinases (nRTKs). Approximately 90 members of the tyrosine kinase family are known, of which 58 are RTKs and 32 are nRTKs [7]. There are 58 RTKs encoded in the human genome, which are further subdivided into 20 subfamilies (Table 1). RTKs belong to the family of cell surface receptors and by being transmembrane proteins crossing the biological membrane, they induce a specific response in the intracellular compartment after ligand binding in the extracellular region [8]. The RTK family includes, for example, epidermal (EGF), fibroblast (FGF) and nerve growth factor (NGF), as well as insulin receptors (IRs) [9,10,11,12]. Numerous nRTKs occurring freely in cells include ABL Proto-Oncogene, FES Proto-Oncogene, Janus Kinase (JAK), Focal Adhesion Kinase (FAK), SRC Proto-Oncogene or Bruton’s Tyrosine Kinase (BTK) [13]. Non-receptor TKs therefore regulate multiple cellular functions ranging from cell proliferation, differentiation, migration and adhesion to apoptosis by fusing with RTKs or other membrane receptor proteins (G protein receptors). Because nRTKs are part of the T- and B-cell activation signaling pathways, they also modulate immune response [11]. Table 1 contains information on receptor tyrosine kinase families, their receptors and basic characteristics. In the following table, we will take a closer look at their structure, activation and subsequent inhibition.

In this review, we offer a thorough overview of the most recent and notable research in the field of protein tyrosine kinase inhibition through the innovative application of nanomedicine. Scientific literature specifically examining the conjugation of metal nanoparticles with tyrosine kinase inhibitors has emerged predominantly within the last decade, with the exception of a study on platinum nanoparticles. Consequently, our review presents a comprehensive and current examination of the key milestones achieved thus far in exploring the use of nanoparticles for inhibiting tyrosine kinase activity. Our review not only examines the unique relationship between nanotechnology and medicine but also offers an overview of specific drugs that have been approved for use in both European and US markets. This consolidation and comparison of the available information stands out as both unique and highly valuable.

## 2. RTK Structure

Receptor TKs (Figure 1) are transmembrane glycoproteins, and all have an analogous structure, which can be divided into an outer, extracellular region to the end of which the ligand binds, the transmembrane helix, and an inner, cytoplasmic intracellular region with a juxtamembrane regulatory region (JMR), a TK domain (TKD), and a terminal carboxyl (C-) tail [35]. The bulk of receptor TKs function as a monomer in an inactive state. The JM region performs important regulatory roles, linking the C-terminus of the transmembrane helix to the intracellular domain and is essential for protein kinase activation. Furthermore, the JM region is significantly involved in the autoinhibition of protein kinase activity by JM regions forming inhibitory contacts with the kinase moiety and this effect has been reported for the Fms-like tyrosine kinase (FLT3), KIT, muscle-specific kinase (MuSK) and Ephrin (Eph) families [2,12]. The kinase domain is composed of an N-terminal lobe, a larger C-terminal lobe, and an ATP binding site that forms the interface between the N- and C-lobes. In addition to this interface between the lobes, there is a C-helix, which is a critical regulatory element, and an A-loop (activation loop), which forms a stretch of the protein substrate binding site, near the ATP binding site. The C-terminal lobe also carries a P-loop (phosphate binding site) and a DFG motif (glutamate, phenylalanine, glycine), which plays an important role in the regulation of kinase activity [36,37].

### 2.1. RTK Activation Mechanism

The recognition of small molecules by proteins is the basis of most processes in the cell. Most enzymes and many components of cell signaling pathways require specific or selective binding of a small molecule or ligand to a protein [38]. RTKs are usually activated by receptor-specific agonists that, when bound to the extracellular region of the RTK, induce dimerization and/or oligomerization of the receptor (this refers to a small subset of RTKs that can form multimers in the absence of the activating ligand) [39]. Binding of the ligand to the extracellular ectodomain facilitates receptor dimerization and induces conformational reorganization in pre-existing dimers. This phenomenon subsequently activates protein kinase activity in the intracellular part of the receptor. Activation of the kinase domain of each monomer in the inner region promotes trans-autophosphorylation of tyrosine residues of the C-terminal tail, the juxtamembrane (JMR) region and possibly also on the activation loop of the monomer opposite—leading to further activation [2]. Increased/abnormal tyrosine kinase activity usually leads to cell proliferation disorders that are closely associated with tumor invasion, metastasis, and tumor angiogenesis. The activation mechanism of receptor tyrosine kinase is shown in Figure 2.

The endothelial growth factors VEGF and the corresponding TK receptors (VEGFR) are crucial in the proliferation of endothelial cells or in the formation and elongation of blood and lymphatic vessels [40]. Another crucial factor involved in tumor cell proliferation and the organization of the tumor microenvironment towards targeted therapy is also the fibroblast growth factor receptor (FGFR). Dysregulation of FGFR in tumors is caused by activating mutations, gene amplifications or chromosomal rearrangements. FGFR aberrations have been identified in 7.1% of tumors, predominantly in urothelial carcinoma, breast cancer and endometrial adenocarcinoma [41].

Genetic changes and abnormalities can alter the activity or regulation of TK functions, and therefore modification of such dysregulated functions is a target in drug development. The drugs developed so far are characterized into two types. The first option is biologic drugs that disable the activation of RTKs either directly or by binding the cognate ligand that is present in vivo at their site of action, and their role is to perform a specific biological function. The second type are small molecules developed to directly inhibit the activity of tyrosine kinases [38,42].

### 2.2. Tyrosine Kinase Inhibitors

Biologically active tyrosine kinase inhibitors have been developed to treat cancer and other diseases driven by changes in receptor tyrosine kinases. Approximately 80 drugs have been approved for use in the treatment of human diseases, with hundreds more under investigation in clinical trials. A common feature of most of the drugs is a significant polypharmacology, which is based on the ability of the ligand to bind to more than one molecular target [43]. This property can increase the effectiveness of drugs, but it can also cause adverse effects beyond their target. However, polypharmacology, with precise planning using advances in molecular sciences and molecular modeling, can become the greatest advantage of these targeted drugs. The result will be a useful additive or synergistic effect on altered biochemical pathways [44]. Previous studies have shown that more than 50% of all proto-oncogenes and oncogenes have tyrosine kinase activity. In addition, abnormal tyrosine kinase activity is also associated with tumor invasion and metastasis, tumor neovascularization, and tumor resistance to chemotherapy [40,45].

These small molecules specifically target the adenosine triphosphate binding site in the intracellular region of the TKD [46]. Inhibition, i.e., the slowing or interruption of a particular process, in the case of tyrosine kinase enzymes, can occur direct binding to the receptor by specific molecules that compete with ATP molecules for a binding site on the tyrosine amino acid residue. By stealing the binding site from the inhibitor, tyrosine kinase phosphorylation is reduced, and the consequence is a slowing (inhibition) of cancer cell proliferation [47].

Small molecule inhibitors attacking the kinase domain are therefore divided into ATP-competitive (type I and II inhibitors) and non-ATP competitive (type III and IV, VI inhibitors). Variable or bivalent inhibitors binding to two different regions of the kinase cube include type V. Competitive inhibitors of type I kinases simulate/imitate the purine ring of the adenine moiety of ATP, and their binding changes the active conformation of the kinase, thereby preventing the transfer of the phosphate group. These inhibitors are formed by a heterocyclic ring that occupies the purine binding site. Type II inhibitors target the inactive form of the DFG-Asp enzyme and form reversible interactions with target kinases via hydrogen bonds in the lipophilic hinge region of the protein. These lipophilic interactions result in the high specificity of type II inhibitors, which is likely reflected in a reduction in toxicity compared to type I.

ATP-competitive TKIs are further divided into reversible (Type I-V) and irreversible. Reversible TKIs include erlotinib and gefitinib which form non-covalent interactions with the kinase residue and their effect can be stopped by the addition of ATP. ATP-competitive irreversible TKIs, on the other hand, form covalent bonds with the kinase domain and the plot is therefore irreversible (osimertinib). Non-ATP competitive type III inhibitors or so-called allosteric inhibitors bind to a site adjacent to the ATP binding site and therefore both ATP and allosteric inhibitors can bind to the protein at the same time. Type IV inhibitors target more distant regions outside the ATP binding site and are substrate competitive. Non-ATP-competitive and irreversible inhibitors include type VI inhibitors that covalently bind to their protein kinase target. For example, the drugs afatinib or ibrutinib form a covalent bond with their target [8,48,49].

There are currently 80 therapeutic agents approved by the Food and Drug Administration (FDA) targeting approximately twenty different protein kinases, with 6 drugs approved in 2023. Of the total, 69 drugs are used to treat cancer, and 6 drugs are used to treat inflammatory diseases (atopic dermatitis, rheumatoid arthritis, psoriasis, alopecia areata, and ulcerative colitis). Except for netarsudil, temsirolimus and trilaciclib, all drugs are taken orally. Twenty-one drugs block non-receptor protein tyrosine kinases and 42 inhibit receptor protein kinases [50]. Figure 3 presents a timeline of medicines approved by the EMA and the FDA from 2001 to the present. The names of approved tyrosine kinase inhibitors (receptor and non-receptor), years of approval for the U.S. and European markets, and additional information are listed in Table 2 and Table 3.

### 2.3. Properties of the TKIs

Tyrosine kinase inhibitors are characterized by low solubility and therefore highly variable bioavailability, which is reflected by different rates of absorption in the gastrointestinal tract. TKIs are well soluble in acidic environments, and their solubility decreases rapidly above pH values of 4–6. TKIs, which are lipophilic and nonpolarized at physiological pH, dissolve freely in lipids and because of the simple diffuse passively across the lipid bilayer of the membrane. In contrast, polarized TKIs are commonly transported by proteins that form transmembrane channels. The effect of some TKIs may be affected by food. Rapid buffering of gastric acid may have a negative effect on drug solubility, whereas for other TKIs, when the drug is ingested with food, there is increased solubility and therefore better absorption and reduced time of the drug in the stomach. This effect may result in a reduction in adverse effects on the gastrointestinal tract [9,54].

All medications (with three exceptions) are taken orally with a strictly defined dosage. This is because when administered intravenously, 90% of TKIs are highly bound to plasma proteins (most commonly α (1)-acid glycoprotein and/or albumin), leading to ineffectiveness of the inhibitors. For example, erlotinib is given at a dose of 150 mg/day, sorafenib is taken at a dose of 80 mg/per day, and sunitinib is given at a starting dose of 50 mg/per day for 4 weeks followed by two weeks of drug omission. Imatinib is the only TKI with nearly 100% absolute bioavailability. Lower absolute bioavailability is predicted for sunitinib and sorafenib (50%) and nilotinib (31%) [55,56]. TKIs are also characterized by relatively long plasma half-lives (mean terminal elimination halftime t_1/2_), ranging from shorter time periods (dasatinib, 3 to 5 h) to long ones (sunitinib, 40 to 60 h; bosutinib, 32 to 39 h), for which once-daily dosing is supported [57,58].

The pharmacokinetics of TKIs should also be considered in multi-drug interactions (DDIs), due to induction or inhibition of metabolic pathways used by TKIs. Drug–drug interactions can lead to fatal consequences, which may result from reduced therapeutic efficacy or severe side effects. The DDI rates ranged from approximately 20% to almost 60% for interactions that reduced the effect of TKIs and up to 74% for interactions that increased the toxicity of TKIs. Pharmacokinetic interactions involve changes in the absorption, distribution, metabolism or excretion of the drug. All these aspects affect the availability of the drug at receptor sites [59].

### 2.4. Adverse Effects

Evidence of possible cardiotoxicity of varying severity has been demonstrated in evaluations of TKIs. For example, dasatinib has been associated in clinical trials primarily with the occurrence of fluid retention, QT interval prolongation (also lapatinib) or pulmonary hypertension; sorafenib has shown an increased risk of hypertension and myocardial infarction [8,60]. TKIs can negatively affect vascular endothelial cells and cardiomyocytes (cardiac postmitotic cells), and vascular endothelial growth factors (VEGFs), which are involved in angiogenesis or myocardial perfusion, are often implicated in target organ damage [61]. A common side effect of anti-VEGF therapy is hypertension, which ranges from 25% with sorafenib, sunitinib, and vandetanib and up to 40% with pazopanib and axitinib. The mechanism of hypertension is not yet fully understood, but studies suggest that single genetic polymorphisms in the genes encoding VEGF-A and its main receptor VEGFR-2 predispose to the development of hypertension [62].

One of the serious side effects of TKIs is ocular toxicity involving corneal ulceration and blindness. Inhibitors belonging to the EGFR family can cause vision-related side effects such as corneal thinning and erosion [63]. EGFR TKIs are also associated with specific adverse effects, the most common being diarrhea, mucositis, rash, and paronychia. In clinical trials, dermatological adverse effects in the form of rash occurred in almost 90% of cases. Inflammation of the mouth (stomatitis) and inflammation of the gastrointestinal tract (mucositis) occurred in approximately 72% of patients in a study in which patients took afatinib. Other side-effects caused by afatinib and by osimertinib were also observed, namely paronychia, an infection of the tissue at the site of the skin-nail interface. Paronychia was experienced by about 56% of patients [64].

Skin tissue such as the epidermis, sebaceous and eccrine glands and dendritic cells are the site of widespread expression of EGFR, which plays an indispensable role in skin development and physiology. Skin toxicity includes, for example, skin fissures, xerosis, rash acneiform or paronychia. Severe skin toxicity reduces quality of life, can cause psychological and physiological problems and may affect adherence to treatment and its subsequent efficacy [65]. Treatment with cabozantinib may cause hair and/or skin depigmentation after approximately 11 weeks and patients may experience photosensitivity with prolonged treatment. Hair and skin respond similarly to treatment with sunitinib and imatinib, with this change being reversible after treatment is stopped. With cyclic sunitinib treatment, alternating horizontal bands of depigmented and normal hair occur. Cabozantinib, imatinib and sunitinib inhibit c-KIT, which regulates the formation, migration and survival of pigment cells (melanocytes). Dry skin, i.e., xerosis, has been reported in association with the use of sorafenib and sunitinib [66]. Observation of the non-target effects of TKIs has also revealed changes in bone metabolism resulting, for example, from long-term use of imatinib. Multikinase inhibitors for the treatment of CML or thyroid cancer may also have off-target effects on bone metabolism [67].

### 2.5. Resistance

In addition to abnormal activation of protein kinase-related signaling pathways, the tumor microenvironment and its metabolism, epigenetic modifications, or resistance to cell death are also involved in tumor development and resistance to TKIs. Reactive oxygen species (ROS) also contribute to the development of cancer cell resistance to TKIs, which can induce oxidative stress and damage cells when concentrations increase above physiological thresholds. Cells protect themselves from damage by countermeasures to compensate for elevated ROS levels and by activating these antioxidant pathways contribute to drug resistance [11,68]. Over 90% of resistance to tyrosine kinase inhibitors is due to acquired mechanisms, such as secondary mutations, bypass activation, or histological transformation. These mutations can prevent TKIs from binding to the kinase, reducing the drug’s effectiveness against tumor cells. Newer TKI drugs are designed to target these specific mutations. Bypass signal activation is a critical factor in developing resistance to TKIs. Despite the fact that TKIs inhibit signaling pathways that control genes, key signaling pathways in tumor cells can still be permanently activated through bypass mechanisms such as MET or EGFR gene amplification, and AXL activation. This continuous activation allows tumor cells to maintain their growth and proliferation, leading to resistance against TKIs. To address this, an effective treatment strategy for overcoming resistance involves simultaneously inhibiting both the activated bypass signals and the original signaling pathways associated with the controlling genes. However, the mechanisms underlying resistance due to bypass activation are complex, and currently, there is no standard clinical treatment for this type of resistance [69].

The therapeutic responses of individual patients receiving TKIs vary depending on factors such as the efficacy and selectivity of TKIs, tumor biology including tumor heterogeneity, and the tumor microenvironment. Drug resistance to treatment with TKIs also develops naturally and can be divided into on-target mutations (EGFR-T790M to first/second generation EGFR-TKIs), off-target mutations (overexpression of HGF, EGFR-independent resistant mechanisms), and histological transformation to a different tumor type (neuroendocrine or mesenchymal). The T790M mutation was positive in 43–50% of patients who were resistant to gefitinib or erlotinib in the treatment of NSCLC. The cause of resistance may be due to a mutation-induced change in the product from threonine to methionine, which represents a steric hindrance. This affects the formation of hydrogen bonds between tyrosine kinases and inhibitors, leading to an inability to bind the inhibitor. In addition to higher selectivity and potency, new generations of TKIs should be capable of better penetration across the BBB and provide safe combinable treatment with other inhibitors or antibodies that, in conjunction, would enhance the efficacy of the TKIs [47,70].

The blood–brain barrier (BBB) is a neuroprotective boundary composed of a single layer of endothelial, ependymal and tanycytic cells. These cells are interconnected together by adherent junctions and tight junctions, which are mainly formed by the proteins occludin and claudin. Phosphorylation of protein kinases serine, threonine and tyrosine is regulated by occludin. The tight junctions of the cells prevent the entry of harmful substances into the brain, and the BBB also slows the entry of drugs into the tumor. The function of the BBB and the corresponding density of the cell monolayer remains unchanged even during changes at the tumor site; therefore, small molecules, including drugs, can be secreted by efflux pumps such as P-glycoprotein (P-gp, ABCB1) or breast cancer resistance protein (BCRP, ABCG2). Examples of TKIs that are removed from the brain by efflux transporters P-gp and ABCG2 or have limited penetration into the brain are erlotinib, gefitinib or neratinib. Osimertinib, afatinib, which are P-gp substrates, or the dual HER1/HER2 inhibitor lapatinib are also washed back into the bloodstream [71].

The successful results of trials that combine chemotherapeutic drugs for various types of cancer have traditionally been explained by the non-overlapping mechanisms of resistance that these drugs exhibit. However, recent insights suggest that the effectiveness of these combinations is due not only to their individual properties but also to their collective effects. In many cases, the combination of two drugs is more effective than either drug used alone. Combination therapies, some of which are already in clinical use, hold significant promise for reducing both intrinsic resistance and the development of drug resistance. An example is the combination of lenvatinib and gefitinib, which has been found to be safe and potentially effective for patients with lenvatinib-resistant hepatocellular carcinoma [72]. Another form of combination therapy could involve merging nanomedicine with immunotherapy. Tumor immunotherapy relies on the ability of immune cells to recognize tumor cells in their early stages and eliminate them by secreting interferon-γ. This approach has become an important strategy for cancer treatment. As a result, research in immunotherapy is focused on understanding the communication between cancer cells and the host immune system [73]. Immunotherapy has become a standard treatment for cancer, but it has limited effectiveness against tumors and can cause systemic immunotoxicity. To address these challenges, researchers are developing nanocarriers that can selectively and efficiently deliver cancer vaccines or immune system checkpoint inhibitors directly to lymph nodes and tumors, maximizing their therapeutic potential. However, the clinical application of these nanocarriers faces obstacles, primarily concerning toxicity and the complexities of producing these drug delivery systems. Reutilizing already approved nanocarriers with established safety records and clinical production could be a promising strategy, presenting a significant opportunity for advancements in immuno-oncology [74].

## 3. Nanomedicine for Cancer Therapy

Nanomedicine involves the application of complex and various nanotechnology approaches to address medical questions and has the potential to significantly improve the diagnosis and treatment of diseases, including cancer. The main goal of nanomedicine in cancer treatment is to ensure drug delivery to the desired target—tumor cells—at the appropriate concentration, with minimal loss of their volume or activity in the bloodstream [9,75]. Standard chemotherapy uses drugs such as doxorubicin, cisplatin or gemcitabine, which, due to their small size, can have relatively poor biodistribution and associated short half-life in the blood, as well as significant accumulation in healthy organs, i.e., outside their primary target. Their unfavorable pharmacokinetics therefore cause serious side effects such as myelosuppression, neurotoxicity, nausea or alopecia. Increasing the systemic size of the anticancer drug to an average of 5–10 nm, i.e., to a value exceeding the renal clearance threshold (~40,000 Da), can prolong the half-life in the blood and improve accumulation at the target site. An example is the encapsulation of doxorubicin into liposomes, which resulted in an increase in plasma half-life from 5 to 10 min (drug alone) to 2 to 3 days (encapsulated drug). A hidden polymer often used in other nanomedicine applications is polyethylene glycol (PEG), which reduces aggregation and opsonization with plasma proteins, contributing to prolonged circulating half-life and reducing cardiotoxicity of the drug-delivery system. Opsonization is the process of absorption of blood protein opsonin onto the surface of a drug or particle and the drug/particle thus marked is subsequently recognized by macrophages and removed from the circulation. This prevents the drug from traveling into cancerous tissue. Thus, PEG shielded liposomal doxorubicin (Doxil) is clinically approved for the treatment of ovarian and breast cancer. The balance between activity and toxicity of the drug is provided by blood circulation and clearance [76,77,78,79].

Accurate delivery of the drug to tumor tissues without inducing adverse effects involves **active** and **passive** targeting. By active targeting of nanoparticles, we mean the formation of specific bindings of modified ligands on the surface of nanoparticles with receptors on the surface of tumor cells. Probably due to their limited targeting efficiency and potential off-target drug delivery, nanoparticles using active transport have not progressed beyond clinical trials. In nanomedicine, passive targeting refers to the administration of nanomaterials intravenously, without specific modifications of the nanoparticle surface, and their spontaneous accumulation in areas of solid tumors with leaky vasculature. Spontaneous accumulation and retention of particles together form the basis of the EPR effect (enhanced permeability and retention effect). The EPR effect is used as a passive carrier target because the tumor vasculature contains openings and poor lymphatic drainage not found in healthy tissues and the nanoparticles are highly permeable due to the pores in the tumor vasculature. In the case of passive targeting, the link between the physical properties of nanoparticles and biological behavior, such as circulation time and accumulation in organs, must be considered in the design of carriers [75,80,81,82,83].

In addition to their favorable structural properties, biocompatibility, and biodegradability, the development of nanocarriers has increasingly focused on issues related to drug delivery and the avoidance of side effects. As a result, there is significant attention on nanocarriers that can regulate factors such as pH, temperature, ultrasonic sensitivity, or enzyme activity, offering the promise of more precise drug delivery [84]. All these specific properties are common in the tumor microenvironment, and it is the acidic pH, caused by increased glycolysis, lack of oxygen, or heightened activity of certain enzymes, that distinguishes tumor tissue from healthy tissue. The development of these types of nanocarriers, particularly polymer-based or hybrid nanocarriers, holds great potential. Advanced designs include biomimetic nanoparticles like exosomes, which play a crucial role as signaling mediators in regulating the tumor microenvironment [84,85]. This method enables selective drug release in response to the conditions within the tumor microenvironment, ultimately improving treatment efficacy and reducing the occurrence of side effects. Not only do they enhance therapeutic efficacy, but they are also utilized in imaging, thereby connecting diagnostics with therapy [84]. This research area encounters significant challenges, especially regarding the biosafety of nanomaterials, their interactions with biological tissues, and their long-term deposition. These factors are vital for designing effective nanocarriers. Monitoring the sensitivity of nanocarriers must prevent the possible inaccurate distribution of stimuli, for example, in the case of acidic pH, which can occur even in a healthy cell. Understanding the biodegradation and toxicity of nanomaterials in the body is essential, but these questions are still in the early research stages [86].

### 3.1. Nanoparticles Characteristics

Nanoparticles (NPs) with the potential for use in nanomedicine must have, in addition to the properties such as biocompatibility, stability, protection of the drug from degradation, and drug delivery to the target site, characteristics that are directly related to their physicochemical properties. The most important characteristics are particle size, particle shape and particle surface. It is reported that nanoparticles with sizes up to 200 nm can cross the BBB via clathrin-mediated endocytosis. However, the size distribution of nanoparticles is not so clear-cut, because on the one hand, NPs should be small enough to escape from macrophages of the reticuloendothelial system, and on the other hand, they should be large enough to prevent extravasation from normal blood vessels. Another factor to consider for size, for example, is that if nanoparticles are used in cancer therapy, the nanoparticles should be as small as possible to allow penetration into the tumor. But conversely, a longer circulation time is directly proportional to the size of the nanoparticles. The shape of the nanoparticles determines whether they will be taken up by the reticuloendothelial system. It is generally considered that the non-spherical shape of NPs is preferable, and this assumption is already inferred from existing biological systems. For example, viruses and bacteria evade the immune response by evolving into non-spherical forms. Thus, nanomedicine takes advantage of already known knowledge and therefore the design of nanosystems favors non-spherical forms that, like viruses or bacteria, would evade recognition by the immune system. For example, gold nanodots have been found to diffuse faster in tumor interstitial fluid than spherical nanoparticles. Or that PEG gold nanorods resided in tumors, whereas gold nanospheres or nanodiscs distributed on their surface. An indispensable characteristic of nanoparticles is their surface area, which greatly influences the half-life of nanoparticles and their action in the bloodstream. Opsonization often occurs when the surface of nanocarriers is hydrophobic or not modified at all. Therefore, opsonization can be minimized by surface modification of the particles with hydrophilic materials such as polyethylene glycol (PEG) or by preparation of a hydrophilic coating using colloidal silica [87,88,89,90,91]. Another option is to coat the surface of the nanoparticles with dysopsonin albumin, thereby reducing recognition by the immune system and removal from the circulation, while increasing the circulation time in the blood and reducing toxicity. On the one hand, modifying the solubility of hydrophobic drugs will improve their dissolution in aqueous fluids, thereby increasing their mobility in body fluids; however, these drugs are not quite able to cross the lipid barrier (BBB). Conjugation of nanoparticles with hydrophilic biomolecules (proteins, peptides) can carry potent anticancer drugs and selectively deliver them to cells requiring treatment. Conversely, conjugation with lipophilic molecules and encapsulation in lipid vesicles improves membrane permeation and are mostly described for oral drug delivery or to overcome BBB [92,93,94,95]. An equally important characteristic of nanoparticles is their surface charge, which plays an important role in interactions at the cellular and molecular level. One of the most used methods to quantify surface charge is to estimate the zeta potential. Nanoparticles with a highly cationic or anionic surface charge absorb more serum proteins in vitro and are thus more permissive to interact with macrophages, in contrast to neutral particles coated with, for example, PEG, which absorb less serum proteins. The surface charge of nanoparticles is strongly involved in electrostatic interactions between nanoparticles and the cell membrane and in the adsorption of proteins onto their surface [96,97].

### 3.2. Clearance Properties

Three main mechanisms of nanoparticle clearance are known: renal, hepatobiliary and via the mononuclear phagocyte system (MPS, also known as the reticuloendothelial system). After intravenous administration, nanoparticles are distributed through the vascular system to organs and peripheral tissues, where they encounter blood cells, platelets, coagulation factors and plasma proteins. Whether absorption or opsonization by serum proteins occurs depends mainly on the size of the particles. In the case of either absorption or opsonization of a nanoparticle by serum proteins, the size of the nanoparticle is greatly increased and the particle diameter thus increasing is referred to as the hydrodynamic diameter (HD), which in turn affects the clearance in the blood. Renal clearance is a complex process that involves glomerular filtration, tubular secretion and subsequent elimination of molecules by urine. Molecules with HD of less than 6 nm are small enough for free filtration, which does not depend on their charge. Glomerular filtration of particles from 6 to 8 nm is already dependent on the charge of the particles, and since the GBM is negatively charged, positively charged particles are filtered faster than equally sized negatively charged particles. Particles larger than 8 nm are no longer subject to glomerular filtration. For example, quantum dots larger than 8 nm are taken up in the reticuloendothelial system (RES) and in the lungs. Filtered molecules can be resorbed from the tubular fluid, and molecules that were not previously filtered can be actively secreted into the lumen of the proximal tubule. The second route is hepatic clearance, which is much more complex than renal clearance. Hepatic clearance represents the primary route of excretion for particles that are not subject to clearance via the kidneys. In humans, the pore size between sinusoidal endothelial cells is approximately 180 nm, which allows hepatocytes to take up and excrete nanoparticles into the digestive system [97,98,99]. It is also known that parenchymal hepatocytes make up most cells in the liver, whereas nanoparticles are usually taken up by non-parenchymal cells (e.g., Kupffer cells). Particles interacting with hepatocytes are eliminated from the body via the hepatobiliary route, followed by their enzymatic degradation and excretion into the bile. For particles to progress successfully into the biliary system, they must first avoid clearance through the MPS. This system removes nanoparticles from the blood by phagocytosing cells in the blood and tissues. These are cells in the spleen, bone marrow or just Kupffer cells in the liver. In the process of phagocytosis, the nanoparticles undergo degradation inside the MPS cells and, if they are not degraded by these intracellular processes, they remain isolated in the spleen and liver for more than 6 months. Kupffer cells express receptors on their surface that can efficiently bind negatively charged nanoparticles; in contrast, hepatocytes more readily accept positively charged particles [97,100]. It has been shown, for example, that intravenously administered gold nanoparticles are primarily removed from circulation by Kupffer cells in the liver, and that the overall removal of nanoparticles is very lengthy. Nanoparticles tend to aggregate into vesicles over time, reducing the surface area of the nanoparticles, which could minimize possible adverse effects [101].

## 4. Nanoparticles in Cancer Treatment

Nanoparticles are materials whose overall dimensions are no larger than 100 nm and are referred to as zero-dimensional because all their dimensions are at the nanoscale, as opposed to, for example, one-dimensional materials with one dimension larger than the nanoscale (nanowires, nanotubes) or two-dimensional materials with two larger dimensions (self-assembled monolayer films). Applications of nanoparticles in medicine are versatile, ranging from contrast agents, to use in imaging techniques, to gene and drug delivery to cells [102]. The investigation of nanoparticles and more advanced nanosystems useful in medicine has already moved from the well-known findings associated with anticancer therapy (improved stability, prolonged circulation time in the body, targeted drug delivery, reduced incidence of side effects, overcoming drug resistance) to the study of the safety and efficacy of these systems. Research focusing on the use of nanoparticles is therefore yielding promising results in vitro and in small animal models. Nevertheless, the number of nanomedicines available to patients is significantly lower. This is due both to the translational gap between animal and human studies, and to the heterogeneity of patients and the biological basis of diseases, which may alter the efficacy of nanoparticles [103,104].

### 4.1. Lipid-Based Nanoparticles

Lipid-based nanoparticles (LNPs) are highly versatile carriers that can encapsulate therapeutics, imaging agents, nucleic acids (DNA, mRNA and siRNA) or monoclonal antibodies and have the advantage of primarily protecting drug substances from in vivo degradation, but also exceptional biocompatibility, increased potency, solubility and the ability to target the drug to the desired site. Already in 1995, Doxil^®^ (liposomal doxorubicin) was approved by the FDA as the first nano-drug with passive targeting for the treatment of solid tumors and, after patent expiration in 2009, its follow-up product Lipodox^®^ was and is still currently approved in the USA. However, Lipodox was rejected by the EMA for the European market due to insufficient evidence of bioequivalence of the free doxorubicin in Lipodox and Caelyx^®^ (the European trade name of Doxil) [105,106,107,108]. In 2012, the FDA approved another liposomal drug, Marqibo^®^ (Vincristine sulfate liposome injection, VSLI), for the treatment of relapsed/refractory ALL (acute lymphoblastic leukemia). Another example is the thermosensitive liposome doxorubicin, Thermodox^®^, which releases the drug when energy is externally applied, making it the first and so far, only drug of its kind. It targets the treatment of colorectal liver metastases and has also shown promising activity in the treatment of chest wall recurrent breast cancer [109,110,111]. In 2019, Khan and his colleagues fabricated lipid-polymer hybrid nanoparticle system composed of a natural polymer of chitosan and soy phospholipid (Lipoid S75) in a lipid-to-polymer ratio of 20:1, and their nanosystem, with an average size of 200 nm, had a loading efficiency of 89.2% for cisplatin. The polymeric region controls the drug release, and the lipid layer prevents drug leakage. Cell viability studies on A2780 cells (ovarian cancer cell line) confirmed the cytotoxic effect and increased cellular uptake of the nanoparticles prepared in this way was demonstrated [112,113].

### 4.2. Polymer-Based Nanoparticles

The uniqueness of polymer-based nanoparticles over other types of nanoparticles for drug delivery lies in the variety of designs of these systems, due to the controllability of size, shape and surface charge. Their sophisticated properties allow them to respond to stimuli associated with specific biological environments and target specific targets in the body. For example, polysaccharides play a significant role in advancing the treatment of colon cancer due to their unique properties. They facilitate the targeted delivery of anticancer drugs directly to the colon, which enhances therapeutic efficacy while minimizing side effects. Numerous polysaccharides, including chitosan, cyclodextrin, pectin, or hyaluronic acid, have emerged as ideal candidates for drug delivery systems because of their biocompatibility, biodegradability, and microbial specificity [114]. As early as 2005, the FDA approved the passively delivered Abraxane^®^ drug, albumin-bound paclitaxel, as a first-line treatment for metastatic breast cancer and advanced non-small cell lung cancer and metastatic pancreatic cancer. It was designed to eliminate the use of the toxic solvent Cremophor (polyethoxylated castor oil), which is usually required when paclitaxel is administered. Paclitaxel particles bound to human serum albumin protein eliminated the need for concomitant administration of potent antihistamines, prevented immunoreaction to Cremophor without the need for dexamethasone administration, and improved infusion time. Compared to Cremophor-based therapy, Abraxane^®^ was found to have increased tumor inhibition due to improved endothelial binding [115,116,117]. Injectable cabotegravir for the treatment of PrEP (pre-exposure prophylaxis) in adolescents and adults was approved by the FDA in 2021 under the brand name Apretude^®^. It is a long active antiretroviral therapy, and this integrase strand transfer inhibitor (INSTI) significantly reduces the risk of HIV-1 infection in higher risk populations. The intramuscular injection suspension contains 600 mg/3 ml (200 mg/mL) of cabotegravir, 105 mg of mannitol, 60 mg of PEG 3350, 60 mg of polysorbate 20, and water [118,119]. Approved treatments for HIV PrEP consist of two oral medications daily (emtricitabine/tenofovir disoproxil fumarate and emtricitabine/tenofovir alafenamide). Cabotegravir injections may provide a benefit for patients with kidney disease or for patients not adhering to oral therapy. Injection administration is by one injection per month initially, with a two-month interval between injections thereafter [120]. As early as 1975, the FDA approved Estrasorb^®^ (estradiol topical emulsion), which encapsulated estradiol hemihydrate using micellar nanoparticle technology. Estrasorb^®^ is the first drug based on this technology to be used for the treatment of moderate to severe vasomotor symptoms (night sweats, hot flashes, flushing) that are caused by menopause [116,121,122].

### 4.3. Metal Nanoparticles

Nanoparticles based on gold, silver or platinum exhibit unique physicochemical properties and have exceptional resistance to oxidation even when exposed to high temperatures. Noble metal nanoparticles have, among other things, special plasmonic properties that allow them to be monitored in the human body at the nanoscale. Such targeted diagnosis and therapy are associated with higher therapy efficacy and lower risk of side effects. As with all research incorporating nanotechnology into medicine, the specific use of noble metals poses a few challenges. On the one hand, their use is extremely important, for example, in precise imaging of the actual stage of tumors, in photothermal and photodynamic therapy, in targeted drug delivery, in radiotherapy, in antimicrobial and anti-inflammatory effects, or in gene silencing and free radical generation, but efforts still need to be made to eliminate toxicity. It is also essential to ensure that measures are in place to regulate the methods of synthesis, stability, appropriate dosage at the appropriate time and place or accumulation of these substances in the body and minimization of non-target effects [123,124].

The optical property that distinguishes metal nanoparticles from other materials is localized surface plasmon resonance (LSPR). The plasmon represents the collective oscillation of the free charge in the metal and can be thought of as a type of plasma wave. The positive electric charge of the metal is fixed, and the free electron is free to move around the nucleus. An external electric field (e.g., a radiation source) causes the free electrons on the surface of the metal to collectively vibrate, leading to the formation of surface plasmons. By having an electric charge, the electrons generate an electric current when they vibrate, and the resonance of the electric field from the vibration of the electrons with the vibration of the external electric field is called surface plasmon resonance [125].

### 4.4. Platinum Nanoparticles

Platinum nanoparticles are exceptionally biocompatible and their greater cytotoxic effect (e.g., compared to AuNPs) is used in cancer treatment, mainly because the interaction of platinum ions with DNA leads to its replication. Platinum in the form of cis-platinum (cis- [Pt (NH_3_)_2_Cl_2_], Pt (II)) is an important chemotherapeutic agent that has successfully increased survival rates from 10% to 85% in patients with testicular cancer. Despite favorable chemotherapeutic responses, several adverse effects are associated with cis-platinum treatment, including the development of drug resistance, possible ototoxicity, nephrotoxicity, or neurotoxicity. Also, the lack of bioavailability after oral administration of cis platinum leads to the need for intravenous administration [126,127]. Most platinum-related studies address the prevention of resistance to platinum-based chemotherapeutic agents or compare treatment with TKIs and platinum-based chemotherapy, and few papers address the conjugation of TKIs to/into platinum nanoparticles or more advanced systems. In a 2022 study, a team of researchers Kao et al. [128] focused on overcoming intrinsic resistance to platinum-based drugs to minimize toxicity during chemotherapy in patients with head and neck squamous cell carcinoma (HNSCC). They combined afatinib (an EGFR tyrosine kinase inhibitor) with pembrolizumab, and the study met its primary endpoint of objective response rate (ORR) with reasonable toxicity in patients with HNSCC [128]. Ji et al. in 2021 [129] presented a study of screening for non-small cell lung cancer (NSCLC) tumors. For the clinical outcomes of ALK-positive lung cancer patients who were treated with an ALK tyrosine kinase inhibitor or those treated with platinum-based chemotherapy, the median overall survival (OS) differed significantly. For patients treated with an ALK inhibitor, the median OS was 72 months, and for patients treated with platinum-based chemotherapy, the median OS was 12 months. ALK inhibitors have demonstrated superior treatment efficacy in patients with ALK-positive lung cancer compared with conventional chemotherapy [129]. In 2015, Kim et al. [130] reported a study to investigate the clinical activity, safety, and predictive biomarkers of dacomitinib (HER TKI) in patients with recurrent or metastatic esophageal squamous cell carcinoma (R/M-ESCC). Dacomitinib has demonstrated clinical efficacy with manageable toxicity in R/M-ESCC, in contrast to platinum-based chemotherapy, which remains the mainstay treatment [130]. Dolman et al. (2012) [131] prepared a conjugate of the tyrosine kinase inhibitor sunitinib with a platinum-based linker to a lysozyme carrier. Pharmacological activity was evaluated in human kidney proximal tubular (HK-2) cells, and the ability of the conjugate to accumulate in the kidney was monitored in mice. The study confirmed that the lysosomal conjugate with TKI and platinum linker potently inhibited tyrosine kinase activity and rapidly accumulated in the kidney after i.v. administration, providing sustained levels of drug there for up to three days [131].

### 4.5. Silver Nanoparticles

The use of silver nanoparticles in biomedical applications is significant and includes medical diagnostics, detection or protective coating of medical devices [132]. Silver nanoparticles can either be designed to specifically target leaky tumor blood vessels and reduce the clearance rate due to the lack of functional lymphatic vessels, or AgNPs can conjugate with antibodies and bind to antigens present at the tumor site [133]. The use of silver nanoparticles as biosensors has also found widespread application in biomedicine for early detection and drug discovery, in the environment or in food control [134].

A study by Piergies et al. [135] presented erlotinib adsorbed onto silver nanoparticles using SERS. In this work, the researchers focused on studying the chemical properties of the TKI + AgNPs conjugate without targeting the biological effects of the system. Characterization of the nanoparticles and the stability of erlotinib deposited on silver nanoparticles indicated a strong interaction of erlotinib with AgNPs. The silver nanoparticles were found to be stable when mixed with erlotinib solution with a concentration lower than 6.77·10^−6^ M. Also, a significant increase in zeta potential from −51 mV to 7 ± 2 mV was observed at a concentration higher than 6.77·10^−6^ M and at a concentration of 3·10^−5^ M. There is a charge inversion of silver nanoparticles due to adsorption of erlotinib molecules having protonated amine moieties. In the case of SERS characterization, one of the most enhanced bands was shown to be due to stretching vibrations of the triple bond between the carbons [1983 cm^−1^], indicating a strong interaction of this group with the silver nanoparticles [135].

In anticancer therapy focused on the study of protein tyrosine kinases, silver nanoparticles are mainly applied as tools for targeting specific TK receptors for more accurate and faster detection, and work on the conjugation of tyrosine kinase inhibitors with silver nanoparticles is also mentioned. Abdelhafez et al. (2021) [136] addressed the antiproliferative potential of crude extract, different fractions and green synthesized silver nanoparticles from soft corals belonging to the family Nephtheidae, against different tumor lineages. Soft coral metabolites were detected by LC-HR-ESI-MS (Liquid chromatography-high resolution-electrospray ionization-mass spectrometry). The characterized metabolites were further analyzed by molecular docking against EGFR, VEGFR and HER2 (ErbB2) proteins, which are significantly involved in the proliferation and survival of tumor cells. The ability to interact with the active sites of EGFR, HER2 and VEGF implies their likely contribution to the antiproliferative potential of Nephthea sp. as tyrosine kinase inhibitory molecules. An interesting finding was that the antiproliferative potential of the total extract against A549 (lung cancer cell line) and MCF-7 (breast cancer cell line) tumor cells was improved after its encapsulation in biogenic silver nanoparticles. The results obtained provided an alternative option for research and development of anticancer therapy using substances of natural origin [136]. Kundu et al. [137] in their work found that protein-coated silver nanoclusters (AgNCs) on graphene oxide (GO) sheets can act as a drug carrier for imatinib, the first tyrosine kinase inhibitor. The envelope proteins used were human serum albumin (HSA) and bovine serum albumin (BSA). AgNCs/GOs also have great potential for use as contrast agents for X-ray computed tomography (CT) imaging. The adsorption of silver NCs on the GO surface and subsequent internalization into K562 cells (human erythroleukemic cell line) was monitored by fluorescence correlation spectroscopy (FCS). K562 cells were exposed to imatinib, GO and silver NCs in cell proliferation study. Cell viability was evaluated by MTT assay, and the study presented several possibilities that were responsible for reducing the toxic effects of silver nanoparticles. In addition to the size or shape of the nanoparticles, the presence of protein moieties emerged as an important factor in reducing the toxic effects of AgNPs. In addition to the size or shape of the nanoparticles, the presence of protein moieties emerged as an important factor in reducing the toxic effects of AgNPs. According to this study, Ag NCs/GO can be a synergistic drug carrier for imatinib, and the immobilization of imatinib and Ag nanoclusters on GO surface with high adsorption capacity represents a promising choice for the design of drug delivery systems. Upon delivery of the system of Ag NCs and imatinib on the GO surface to K562 cells, a further 8–12% reduction in cell viability was observed compared to the effect of imatinib itself [137].

### 4.6. Gold Nanoparticles

Gold nanoparticles are very popular in medical applications due to their unique functional properties and ease of synthesis. The optical, electronic, physicochemical properties of gold nanoparticles are modifiable by changing their basic characteristics such as size, shape and aspect ratio. The most common medical applications of AuNPs include photothermal (PTT) and photodynamic (PDT) therapy, drug and gene delivery systems, radiotherapy (RT), X-ray imaging, computed tomography (CT) or diagnostics. Gold nanoparticles are increasingly appearing in bone tissue engineering and regenerative medicine applications (TERM) due to their ability to deliver bioactive molecules, enhance stem cell differentiation, and monitor implanted cells. AuNPs can improve the adhesion and mechanical properties of the scaffold and cell–cell interactions [138,139].

Photothermal therapy (PTT) is a non-invasive and selective method of treating many cancers and works through the photothermal effect, which is induced by the conversion of light energy into heat. Gold nanoparticles act as a photothermal agent which, when accumulated in the tumor cell and illuminated by NIR radiation, induces a plasmon resonance on the surface of the nanoparticle, thereby locally raising the temperature in the cell above 42 °C. Healthy cells, unlike tumor cells, are more resistant to higher temperatures and therefore the primary tumor cells are destroyed by the higher temperature, thereby reducing side effects. Also, localized heat generation improves the permeability of the tumor vasculature and thus leaves room for drug delivery or accumulation, for example, in combination therapy with chemotherapy. The near-infrared (NIR) laser is desirable in PTT because of its lower scattering and its absorption in tissues, allowing it to penetrate longer compared to visible light [140,141]. Photodynamic therapy (PDT), like PTD, is a minimally invasive treatment that destroys tumor cells using a photosensitizer (PS), a specific wavelength, and in the presence of oxygen. The accumulated photosensitizers, when activated by light at the target sites, induce the production of reactive oxygen species (ROS), which are cytotoxic and lead to cell death via apoptosis or necrosis. Precise targeting of tumors, minimal toxicity or the availability of repeated treatments are the main advantages of the use of photodynamic therapy; however, PDT is associated with limited penetration of radiation into deeper tissues or organs or the development of skin photosensitivity after treatment. In the case of photodynamic therapy, gold nanoparticles are carriers of photosensitizers, and this combination offers several advantages, including enhanced production of singlet oxygen and other ROS, the possibility of conjugation of biological ligands with PS-AuNPs for active targeted therapy, or the variability in shape and size of gold nanoparticles. Another advantage is the promising combinability of PDT and other therapies such as PTT [142,143].

In one of the most recent studies (May 2024) focusing on the conjugation of gold nanoparticles with a tyrosine kinase function-blocking anticancer drug, Munteanu et al. [144] focused on Pluronic gold nanoparticles (AuNPs-PLU) loaded with the tyrosine kinase inhibitor (TKI) midostaurin (MDS), the first targeted therapy for Fms-like tyrosine kinase 3 (FLT3) mutant acute myeloid leukemia (AML). The preclinical study is comparing the efficacy of AuNPs + MDS and the MDS inhibitor alone in vitro and in vivo models in MV−4–11 Luc2 leukemia cells. Midostaurin bound to nanoparticles showed superior tumor inhibition effect in vivo compared to the free drug and succeeded in sustaining tumor formation without significant growth in the first half of treatment. In vitro cytotoxicity evaluation by XTT assay was performed for the free drug MDS, for AuNPs-PLU alone and for the conjugated AuNPs-MDS-PLU system. Unloaded AuNPs-PLU showed no cytotoxicity, free drug and AuNPs-MDS-PLU inhibited the growth of leukemia cells in a dose-dependent manner, with very low doses of MDS and AuNPs-MDS-PLU reducing cell viability by almost 50%. Although the free drug MDS reduced tumor formation, it failed to stop tumor development. AuNPs-MDS-PLU system was able to significantly minimize tumor growth and an obvious improvement in terms of tumor development was also observed. In vivo testing was performed in a mouse model and control buffer, free drug, nanoparticles and drug-loaded nanoparticles were administered by intraperitoneal injection. There was no significant tumor inhibition in the case of administration of buffer and nanoparticles alone. For example, in previous works, researchers have shown that FLT3 inhibitors on Pluronic gold nanoparticles inhibited tumor growth and had a better therapeutic effect compared to the drug alone; in another study, they prepared gelatin-coated gold nanoparticles with the FLT3 inhibitor bound on the surface and demonstrated an enhanced anti-tumor effect due to increased transmembrane delivery in AML cells [144]. In their work, Zhao et al. [145] developed a nanotherapeutic platform composed of bimetallic gold–silver hollow nanoshells (AuAg HNSs) and the TKIs pyrotinib (PYR) and Herceptin (HCT) [145]. Pyrotinib is a novel irreversible dual EGFR/HER2 tyrosine kinase inhibitor and under the trade name Irene, it is approved so far only in China (as of 2018) for the treatment of metastatic breast cancer [146]. Herceptin with the drug trastuzumab is an antibody-targeted therapy for the treatment of patients with human epidermal growth factor receptor 2-positive (HER2+) early breast cancer (EBC) [147]. The developed platform exhibits multifunctional effects through chemo- and photothermal activity, oxidative stress and immune response. HCT-modified nanoparticles exhibit efficient internalization by cells in vitro assays, and the released TKI pyrotinib inhibits cell proliferation (Figure 4). Photothermal effect and subsequent apoptosis of tumor cells is induced by NIR laser and intracellular ROS trigger ferroptosis of tumor cells. Cytotoxicity was evaluated by a CCK-8 assay on BT474 cell line and PPA (PEG-PEI-AuAg) and PPAH (PEG-PEI-AuAg@HCT) showed cell viability of more than 95%; however, after NIR laser application, PPAH showed significant toxicity to BT474 cells. The antitumor activity of PPAPH (PEG-PEI-AuAg@PYR@HCT) exceeded that of PPA and PPAH at the same doses, which could be explained by the synergistic effect of photothermal ablation and chemotherapy. This multimodal therapy also down-regulates genes related to signaling pathways (TNF and NF-κB) and increases immune activation. The developed nanosystem has also confirmed its effects in in vivo tests, where a significant reduction in tumor volume after treatment was demonstrated [145,146,147].

Lv et al. [148] fabricated a novel multifunctional targeted drug delivery system based on gold nanoparticles with PLGA (poly (lactic-*co*-glycolic acid) core loaded with the EGFR tyrosine kinase inhibitor gefitinib and IR780. The fluorescent dye IR780 can promote the generation of ROS due to ultrasound irradiation, and with the help of low-temperature PTT, it can promote drug release and enhance the cytotoxic effect of ROS. The aim of this work is to present valuable therapy in the treatment of patients with Non-Small Cell Lung Cancer (NSCLC) who are resistant to tyrosine kinase inhibitor therapy. The modified gold nanoshell surface enables a photothermal effect for thermally triggering drug release. Cell viability was assessed by CCK-8 assay on PC-9GR cells (gefitinib-resistant cancer cell line), while the nanoparticles without gefitinib maintained more than 80% viability and the cell survival rate decreased to 56% when the nanoparticle system of cRGD-GIPG NPs (cyclic arginine-glycine-aspartic acid-Gef@IR780@PLGA@Au, 200 μg mL-1) was conjugated with gefitinib (Figure 5). The ability of photothermal conversion at the tumor site was monitored using an infrared thermal imaging camera and laser 8 h after injection of nanoparticles with TKIs into mice. Nanoparticles with gefitinib were able to increase tumor temperature up to 42.8 °C after 5 min of irradiation. Furthermore, the antitumor efficacy was evaluated in PC-9GR tumor-bearing mice, and the best effect was observed using the cRGD-GIPG+laser+ultrasound nanoconjugate, where the tumor tissue was almost eliminated (Figure 5) [148].

Kang et al. [149] prepared gold–titania nanostars (GTNs) as TKI carriers of the drug sorafenib and photosensitizers, useful in the treatment of renal cancer, by simple synthesis. Sorafenib was bound to the GTN surface by non-covalent interaction via isothermal adsorption, and 94.8% of the drug was released within 8 h under dynamic shaking conditions in PBS and under static conditions in water. Further release of sorafenib occurred after local irradiation with a wavelength of 808 nm. To confirm the therapeutic efficacy of monodal, dual and trimodal treatments based on chemo C-, thermo T- and dynamic D- strategies, fluorescently labeled (Cyanine5) peptide ligand loaded onto gold–titania nanostars Cy5-RGD-GTN with sorafenib was used, whereas trimodal treatment with synergistic effects resulted in significant cell death (90.33%). At the same time, preclinical assays and histological analyses after in vivo testing in mice confirmed that trimodal GTN-based therapy of renal cancer with the TKI drug sorafenib effectively suppressed tumor growth and significantly reduced the hepatotoxicity, a typical side effect of sorafenib [149]. Molinari et al. [150] conjugated gold nanoparticles with the pyrazolo [3,4-d] pyrimidine derivative SI306, which is a c-Src inhibitor. Src-family protein tyrosine kinases are indispensable in the initiation of signal transduction through the B-cell antigen receptor (BCR). Pyrazolo [3,4-d] pyrimidine derivatives are a promising class of inhibitors with potent proliferative and proapoptotic effects against neuroblastoma, glioblastoma, and chronic myeloid leukemia. The loading efficiency of AuNPs-SI306 was 65% and the system showed satisfactory stability in polar media and in human plasma. The antitumor activity of the nanosystem was evaluated in vitro in a glioblastoma model and was also combined with radiotherapy (RT). The proliferative activity of the AuNPs-SI306 nanosystem, non-functionalized AuNPs and free SI306, either alone or in combination with radiotherapy, was shown using a cell low-density growth assay on the U87 GBM cell line. AuNPs alone and AuNPs-S306 (1 μM) without RT did not show any toxicity to tumor cells, a slight decrease in viability to a value of approximately 80% was observed with the free drug SI306 (1 μM), a more significant decrease in viability (by approximately 40%) without combination with radiotherapy occurred with AuNPs-SI306 (10 μM) and with the drug SI306 alone at a concentration of 10 μM (a decrease of approximately 50%). With the use of radiotherapy, a rapid decrease in tumor cell viability occurred in all sample types, with the most pronounced decrease in the free drug and AuNPs-SI306 (10 μM) to approximately 45% and in AuNPs-SI306 (10 μM), where cell mortality was nearly 70% [150,151]. In 2019, a study by Coelho et al. [152] was published in which they focused on drug delivery systems based on pegylated gold nanoparticles (PEG AuNPs) conjugated with the doxorubicin and drug varlitinib. Varlitinib as a selective dual inhibitor of ErbB-2 tyrosine kinase (Her-2/neu) and EGFR was developed by Singapore-based Aslan Pharmaceuticals but failed in a mid-stage study in 2019. The cytotoxic effect of nanoconjugates and free anticancer drugs was evaluated by colorimetric assay with the Sulforhodamine B (SRB) assay on hTERT-HPNE, S2-013 and MIA PaCa-2 pancreatic cancer cells. Free varlitinib or varlitinib combined with doxorubicin did not show cytotoxicity in any cell line. S2-013 cell viability decreased to 95% at concentrations of 80–100 nM when treated individually with DoxPEGAuNPs, and S2-013 cell viability decreased to 50% when treated individually with VarlPEGAuNPs and at concentrations of 250–750 nM. In the same concentration range, cell viability decreased to 7–25% after addition of the DoxPEGAuNPs/VarlPEGAuNPs combination [152,153,154]. In their work, Codullo et al. [155] tested the in vitro and in vivo activity of imatinib-coated gold nanoparticle (Im)-loaded AuNPs in SSc-ILD (Scleroderma-Associated Interstitial Lung Disease) cells in an experimental model of pulmonary fibrosis. Lung fibroblasts (LFs) and alveolar macrophages (AMs) from bronchoalveolar lavage fluids of SSc-ILD patients were cultured in the presence of nanoparticles. The modified gold nanoparticles were used as a targeted delivery system that provides enhanced efficacy of antifibrotic drugs. Gold nanoparticles functionalized with anti-CD44 antibody and loaded with imatinib (GNP-HCIm), particles without imatinib (GNP-HC) and free imatinib were evaluated in an in vitro MTT assay in a lung fibrosis model of LFs after 24, 48, 72 and 96 h. GNP-HCIm affected cell viability more significantly than imatinib alone, which had no effect. GNP-HCIm showed efficacy of up to 96 h, indicating a slow release of imatinib from the conjugate, and this was confirmed when comparing the effect of imatinib-free GNS-HC nanoparticles, which did not significantly modulate cell viability at the same concentration (Figure 6). In vivo studies confirmed the results of in vitro testing and showed that intratracheal administration of GNP-HCIm was as effective as IP administration of imatinib. As expected, GNP-HCIm reduced the phosphorylation of c-Abl and PDGFR proteins, indicating the selective activity of the drug delivered by nanoparticles. In vivo experiments demonstrated the tendency of GNP-HCIm to accumulate in AM and therefore these cells were investigated as a therapeutic target for inhaled GNP-HCIm, since depletion of AM can reverse lung fibrosis [155].

In their work, Cryer et al. [156] presented a gold nanoparticle nanoconjugate with afatinib Afb-AuNPs to provide better drug compatibility and improve its efficacy. During in vitro testing using MTT, significant dose-dependent cytotoxicity was observed on PC-9 lung adenocarcinoma cells, with tumor cell viability significantly decreased by approximately 60% at an AuNPs concentration of 1 μg/mL, by approximately 70% at c = 3 μg/mL, and the most significant decrease occurred at c = 10 μg/mL, when tumor cell viability reached values below 20%. The versatility of the system was also demonstrated in the A549 lung adenocarcinoma cell line, in which wild-type EGFR expression is not sensitive to TK inhibitors. Afb-AuNPs retained efficacy at higher concentrations, specifically at c = 10 μg/mL, viability was reduced by 42% and a dose with a concentration of 30 μg/mL reduced tumor cell viability by 60%. Nanosystem biocompatibility was observed when Afb-AuNPs were applied to human alveolar epithelial type I cells (healthy lung epithelium), which retained viability and released fewer pro-inflammatory cytokines [156].

In an earlier publication, Bloom et al. [157] investigated gold nanoparticles functionalized with the TKI nilotinib in the presence of mammalian cells using imaging cluster secondary ion mass spectrometry. It was found that nilotinib and gold nanoparticles are distinguishable after internalization into the cell, probably due to the cleavage of nilotinib from the nanoparticle [157]. The physicochemical properties of TKI vandetanib conjugate and micellar AuNPs (ZD6474-AuNM) functionalized with PEG-PPG-PEG were investigated by Sarkar et al. [158] in their study and they found that both AuNMs and ZD6474-AuNMs with a diameter of about 70 nm exhibited high stability in saline; however, there was slow release of inhibitor from micellar nanoparticles in acidic environment (pH 5.2). In vitro testing on MDA-MB-231 and MDA-MB-468 cells (human breast cancer cell lines) showed that ZD6474-AuNM inhibited the proliferation of tumor cells, also reduced their migration and invasion and conversely induced apoptosis [158]. The in vivo testing outcomes are shown in Figure 7.

Vinhas et al. [159] functionalized gold nanoparticles with single-stranded oligonucleotide DNA that selectively targets the e14a2 BCR-ABL1 transcript expressed by K562 (human erythroleukemic cell line) cells was able to efficiently silence the genes and thereby significantly increased cell death. At the same time, the combination of the silencing nanoconjugate with the TKI imatinib caused a decrease in the IC50 of imatinib and reduced the viability of imatinib-resistant K562 cells [159]. In their work, Suarasan et al. presented gold nanoparticle-based systems conjugated with four different FLT3 inhibitors, namely midostaurin (MDS), sorafenib (SRF), lestaurtinib (LST), and quizartinib (QZR). All TK inhibitors were independently conjugated to gold nanoparticles and coated with gelatin, and their use was intended to enhance the efficacy of acute myeloid leukemia (AML) treatment. The multi-kinase inhibitor lestaurtinib is the only one of these four inhibitors that is not being continued in use or production because it failed as a post-chemotherapy treatment for patients with FLT3-activating mutations in a phase III trial and did not show significant clinical benefit in a separate analysis. In vitro evaluation based on the MTT assay was performed for gelatin@AuNPs, gelatin@AuNPs-QZR and for the free drug QZR on OCI-AML3 and THP-1 (acute myeloid leukemia cells) samples over 1 to 14 days. The most pronounced decrease in viability occurred for both cell types with the gelatin@AuNPs-QZR sample, at 7 days after administration, when cancer cell viability dropped below 50%. The high efficiency of the system occurred mainly due to the gelatin coating of the gold nanoparticles, which effectively protects the drug and allows the drug to be administered over several days of incubation [160,161]. Gold nanoparticles modified with short, double stranded DNA oligonucleotide were described in the work of Gossai et al. [162] as a drug delivery system, specifically the TKI dasatinib, which is selectively activated in the tumor cell by the presence of mRNA that is unique to that cancer cell. It has been observed that the amount of drug released from the gold nanoparticle is proportional to the amount of specific mRNA present in the cell. The efficacy of delivery has been validated in vitro and in vivo, and this approach, because of its high customizability with respect to the specific mRNA, can increase drug efficacy and reduce toxicity [162]. The synergistic effect of the TKI sorafenib and gold nanorods encapsulated in HSA using photothermal ablation was investigated in a mouse model of human metastatic clear cell renal cell carcinoma (RCC) in a study by Liu et al. [163]. A similar necrosis outcome was observed in the laser alone (62%) and the HSA-AuNR sample using the laser (63.4%), but the combined treatment of the gold nanoparticle sample with sorafenib and using the HSA-AuNR-SFR laser was the most significant, with 100% necrosis of tumor cells and a significant reduction in tumor size [163].

The study by Lian et al. of the TKI icotinib deposited on the surface of gold nanoparticles, despite it not addressing the effect of the conjugate on biological systems, provides valuable insights and adds excellent information on the mechanism of drug adsorption onto the nanoparticle using techniques such as SERS, UV/vis or DFT. They investigated the charge transfer effect between AuNPs with icotinib molecules and confirmed the adsorption of the drug on AuNPs by UV/vis spectroscopy and transmission electron microscopy (TEM). Icotinib is an EGFR tyrosine kinase inhibitor used for the treatment of non-small cell lung cancer (NSCLC), and its use has been approved by the Chinese National Medical Products Administration (CNMPA) since 2011 in China under the trade name Conmana^®^. Measurement of the absorbance of icotinib conjugated to gold nanoparticles over 30 min confirmed the attenuation of the characteristic peak for AuNPs at 533 nm and there was a shift to higher wavelengths. This suggests charge transfer at the surface of the drug and AuNPs and the formation of an icotinib-AuNPs complex, which stabilized after approximately 20 min by saturation of icotinib from fully charged AuNPs. It was evident from the TEM images that in the case of icotinib-AuNPs, a transparent hydrophobic layer formed around the gold nanoparticles, which is due to the chemisorption of icotinib on the surface of the AuNPs. In addition to UV/vis and TEM, density functional theory (DFT) and surface-enhanced Raman spectroscopy (SERS) were also used to study the adsorption of the TKI icotinib onto the surface of gold nanoparticles. Using molecular electrostatic potential, it was shown that icotinib adsorbed onto the surface of AuNPs via four sites with strong electronegativity, namely near the acetylene group, two N atoms of the quinazoline ring and the O1 atom of tetraoxy cyclododecyl. Comparison of the Raman spectra of icotinib and icotinib-AuNPs showed the phenomenon of selective enhancement of spectral peaks. This charge transfer effect between the drug molecule and the metal increases the polarizability of the molecule and enhances the Raman cross section of the molecule, leading to the enhancement of the peak spectrum. The signal enhancement due to the plasmon effect is much higher than the chemisorption enhancement, while the selectivity enhancement phenomenon cannot be distinguished [164,165]. The above-mentioned studies on the conjugation of TKIs and metal nanoparticles are summarized in Table 4.

## 5. Summary and Prospects

Since the early 21st century, tyrosine kinase inhibitors have become an important treatment option for a wide range of tumor diseases. Selectivity and minimal off-target side effects are the basic prerequisites of an ideal tyrosine kinase inhibitor. Because of the increased safety and human tolerability of the treatment, the suppression of kinase function rather than complete inhibition appears to be an option. This principle is not universal and will depend on the specific type of disease; for example, in inflammatory and autoimmune diseases, an attenuated response may lead to favorable clinical outcomes, whereas in cancer, growth inhibition or suppression is more important [166]. Given the heterogeneity of resistance mechanisms to tyrosine kinase inhibitors and the nature of the tumor, it is not possible to overcome drug resistance in all patients with a single strategy. Precision (or personalized) medicine, improved mutation detection, individualized biological profiles, immunolabeling, proteomics, metabolomics or RNA analysis can attempt to optimize the therapeutic approach [11]. The development of new tyrosine kinase inhibitors may also be aided by artificial intelligence (AI) and machine learning, which are used in molecule design, computer-aided screening or the prediction of protein–protein interactions. Using genomics and computational methods, for example, new targets and drug-resistant mutations have been identified, showing promise in repurposing existing kinase inhibitors or compounds already used to treat cancer or other diseases [167]. The aim of personalized medicine is to avoid a universal approach and instead tailor treatment and prevention interventions precisely to the patient, thereby improving health outcomes, reducing adverse effects of treatment, improving the efficacy of medicines and improving overall quality of life. The incorporation of AI into personalized medicine, which considers the unique individual aspects of each individual, is a significant advancement in the field of personalized medicine. It is a promising tool for more effective treatment and healthcare and can also assist in the analysis of large and complex data, for example, to predict drug response or by matching a treatment plan to a patient’s specific genetic makeup. A specialized subset of AI is generative artificial intelligence, which is indispensable in the development of precision medicine. This innovative field uses machine learning models to generate novel yet realistic outputs and by generating synthetic yet authentic data, generative AI models improve data analysis and interpretation [168,169]. Nanotechnology has huge significance and potential in medicine and anticancer therapies, which should be developed in conjunction with the provision of biosafety that considers potential risks to human health and the environment. Thorough preclinical and clinical studies are essential to evaluate the safety of the use of nanomaterials. Factors influencing biosafety include the basic physicochemical properties of nanomaterials, such as their size, shape, surface charge, chemical composition or also degradability, distribution and biocompatibility [170].

The clinical implementation of nanomaterial-based delivery systems for tyrosine kinase inhibitors presents promising opportunities for enhanced therapeutic efficacy and reduced toxicity. However, several fundamental challenges must be addressed. In addition to concerns about the biological aspects of nanomaterials, particularly their toxicity and safety, there are significant regulatory hurdles. This is primarily due to the absence of specific regulatory guidelines for the development and characterization of nanomaterials. The structure-function relationships of various nanomaterials, along with their properties, compositions, and surface treatments, play a crucial role in their interactions with biological systems. Evaluating all potential effects of nanomedicines in the body is crucial, particularly regarding the formation of aggregates. These aggregates can display different dissolution properties and may result in unexpected toxic effects [171]. The future development of nanomedicine relies heavily on advancements in manufacturing technologies and, crucially, on strong collaboration among academia, industry, and regulatory authorities. Additionally, the acceptance of nanomedicines will be shaped by reliable clinical data and safety studies. The growth of personalized medicine depends on thorough clinical validation of treatments. For these treatments to transition from research to clinical practice, it is essential to ensure that results are consistent across various patient groups and different healthcare settings [172]. In conclusion, it is important to consider how nanocarriers interact with the immune system during the development of nanomedicines for cancer treatment, as these two areas are closely linked. The use of nanoparticles in conjunction with anticancer drugs may produce different effects on the host immune response compared to free drugs, and this response can occur either spontaneously or intentionally. While there are concerns regarding allergic reactions to nanomaterial-based drugs, existing research suggests that a targeted approach could have a positive impact on the immune system. This might include effects on immunogenicity, interactions within the tumor microenvironment, and overall tolerability [173].

In this review we have summarized the most important recent work focused on mechanisms of detection or inhibition of protein tyrosine kinases using modern nanomedicine. A common feature of all studies focusing on the conjugation of tyrosine kinase inhibitors to the surface or inside nanosystems is the specific targeting of the drug to the cells, ensuring a longer circulation of the drug in the circulation, reducing toxicity and, with precise and targeted treatment, also reducing adverse effects. Aspects that must not be neglected in the research of nanomaterials in anticancer therapy are safety, toxicity, accumulation of nanoparticles in the body or naturally evolved resistance. Of the selected studies addressed in this review, the most notable and influential contribution is in the research on gold nanoparticles as nanocarriers. A considerably smaller amount of work is devoted to silver and platinum nanoparticles, whose main drawback is probably their higher cytotoxicity. The studies focusing on nanosilver have mainly used the metal for analytical purposes, i.e., for more accurate and faster detection of metabolites using the SERS method, or as contrast agents in CT imaging. With these methods, a better understanding of the mechanism of tumor tissue formation is possible, as well as possible early detection and diagnosis. Work on nanoplatinum has focused primarily on the prevention of resistance to platinum-based therapeutics and has investigated how tyrosine kinase inhibitors in conjunction with conventional chemotherapeutic drugs can overcome the barriers associated with resistance and toxicity during the chemotherapy treatment. Gold nanoparticles have become a very popular and frequently used inorganic carrier in nanomedicine due to their almost unlimited tunability in size, shape and functionalization from DNA, peptides to polymers. By being able to influence the basic physicochemical properties of gold nanoparticles, such as size and shape, gold nanoparticles can, in simplistic terms, be prepared precisely tailored to the desired application. The size of the nanoparticles influences their ability to cross the BBB or whether or not they will be recognized by macrophages. The functionalization of the gold nanoparticles is of huge importance, firstly to help protect the drug bound to/on the nanoparticles and the second important factor is to adapt to the body’s environment so that the gold nanocarrier is delivered to the target site without being trapped by the reticuloendothelial system. It is known that some tyrosine kinase inhibitor anticancer drugs have low absorption and only a small percentage of the ingested drug will be transported to the target site of action. The conjugation of gold nanoparticles and tyrosine kinase inhibitors is therefore very substantial, combining both the exquisite properties of gold nanoparticles and the therapeutic effects of selective TKIs. Studies clearly agree that the main benefits of the combination of Au nanoparticles and TKIs are more precise and controlled drug targeting during therapy, a greater amount of drug delivered to the target site, and the associated increased efficacy of treatment. By delivering the drug precisely to the intended site, this significantly reduces side effects, which in many disease conditions can be more challenging than the therapy itself. Another huge advantage of gold-based nanocarriers is also the ability to release the drug gradually based on specific conditions such as pH or temperature. In general, gold has lower cytotoxicity and higher biocompatibility compared to other metallic nanoparticles and therefore its use in medicine for the design of new drug delivery systems is extremely important. The unique optical properties of gold nanoparticles are predisposed for use in photothermal therapy, where a local temperature increase due to the plasmon effect occurs after irradiation of a nanoparticle with a wavelength in the near-infrared region. In a system where the gold nanoparticle with the bound drug is located in the tumor tissue, the locally elevated temperature can destroy the tumor cells and spare healthy cells in the surrounding area, which have a higher thermal sensitivity. In this case, there may be a synergistic effect of the gold nanocarrier and its plasmon resonance capability, as well as the effect of the tyrosine kinase inhibitor therapeutic itself. This type of treatment is highly selective and is of great importance in terms of reducing treatment side effects and increasing treatment success. Of the selected studies in this review, the use of conjugates of TKIs with metal nanoparticles in photothermal therapy has not received much attention. For example, one of the papers was a study devoted to the treatment options for TKI therapy-resistant NSCLS patients, which described the use of low-temperature photothermal therapy to facilitate drug release. The utility of metal nanoparticles with bound tyrosine kinase inhibitors, in association with photothermal therapy, can enhance the effect of the drug itself, targeting the specific site of action and thus contributing to better therapeutic outcomes. This area of research provides a broad field of application for metal nanoparticles and their characteristic in the induction of photothermal effect, offering us countless opportunities to explore important metal-based nanosystems with specific drugs for targeted therapy.

The studies on metal nanoparticles in conjugation with TKIs are still at the beginning of their journey and are so far limited to in vitro and in vivo testing, due to the most important issue, which is the human safety of using nanoparticles in the treatment of cancerous diseases in a clinical practice. There are countless known advantages by which even metal nanoparticles have found their place in nanomedicine, but there are still several precisely undescribed and misunderstood mechanisms that hinder the use of these systems in clinical trials. Similarly, in vivo research and preclinical studies, which are an integral part of the process of investigating the efficacy and safety of newly developed drugs, are lacking. There is no doubt that this line of research is of great importance and can significantly support recent conventional cancer treatments, in addition to entirely new therapies based on advanced drug design.

## Figures and Tables

**Figure 1 pharmaceutics-17-00783-f001:**
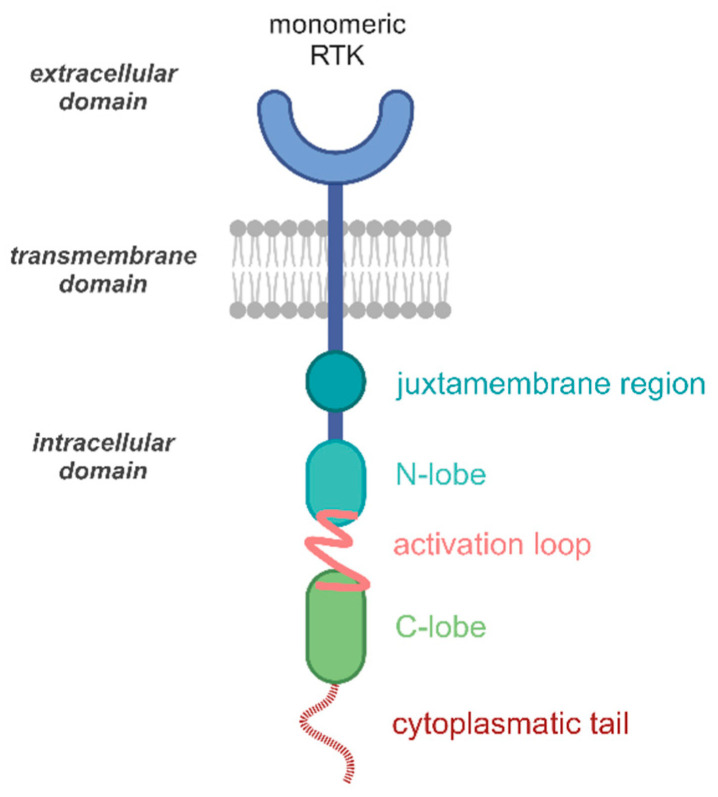
Structure of receptor tyrosine kinase. Receptor tyrosine kinase is composed of an outer and an inner part that are joined by the cell membrane. The extracellular part contains the ligand binding site, and the internal, cytoplasmic part is formed by the juxtamembrane region and TK domain with an N-lobe, an activation loop, a C-lobe, and a tail end with a phosphate binding site. The ATP binding site forms the interface between the N- and C-lobe. Created in BioRender.

**Figure 2 pharmaceutics-17-00783-f002:**
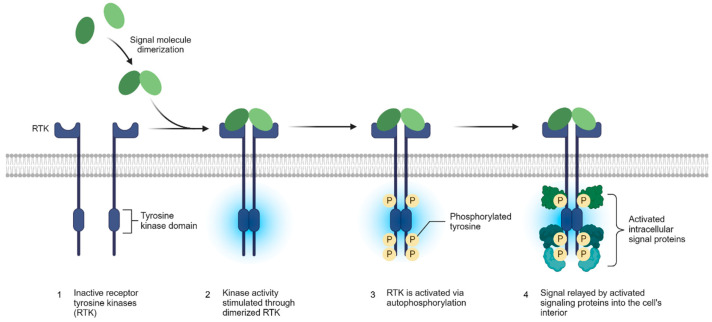
Activation of receptor tyrosine kinase. 1 The inactive RTK monomer. 2 Receptor kinase activation begins with ligand binding to the extracellular portion of the receptor. This leads to activation and subsequent dimerization of the receptor. 3 The response is activation of protein kinase activity in the cytoplasmic portion of the receptor. 4 Activation of TKD in the intrinsic region promotes autophosphorylation of tyrosine residues at the C-terminal tail, the JMR, and at the activation loop and leads to further activation. Abnormal tyrosine kinase activity usually causes cell proliferation disorders. Created in BioRender.

**Figure 3 pharmaceutics-17-00783-f003:**
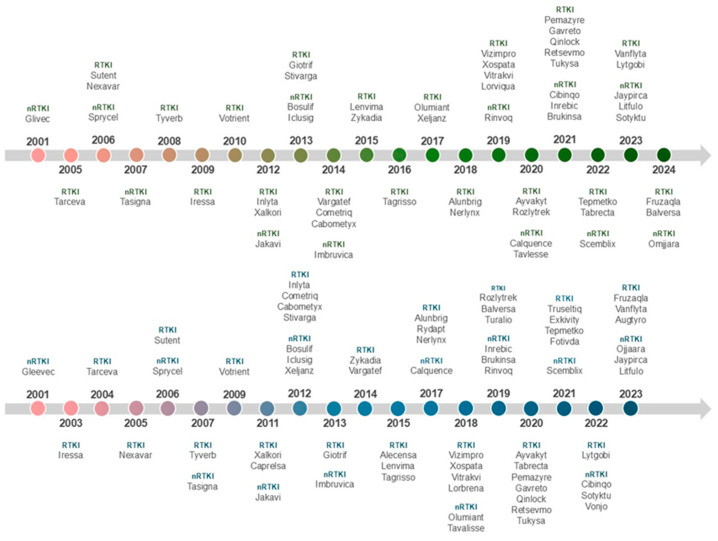
Timeline of EMA (top) and FDA (bottom) approved TKIs. The first approved tyrosine kinase inhibitor was a drug named Glivec (Europe)/Gleevec (US) in 2001, which targets the non-receptor tyrosine kinase BCR-ABL. Active substance imatinib is intended for the treatment of myeloproliferative disorders. The first TKI targeting receptor tyrosine kinase was Iressa with the therapeutic substance gefitinib (EGFR primary target), which was approved by the FDA in 2003. The EMA approved the drug under the same name in 2009. Iressa is used for the treatment of non-small-cell lung carcinoma. In 2023, the FDA approved a total of 6 new drugs directed to treat acute myeloid leukemia (Vanflyta), non-small-cell lung carcinoma (Augtyro; disapproved by EMA), mantle cell lymphoma (Jaypirca) or alopecia areata (Litfulo). Currently, the latest approved drugs in the EU are Fruzaqla (Fruguintinib) for the treatment of metastatic colorectal cancer, Balversa (Erdafitinib) for the treatment of urothelial bladder and urinary cancer, which was already approved by the FDA in 2019, and one non-receptor inhibitor Omjjara (US Ojjaara) for the treatment of myeloproliferative diseases.

**Figure 4 pharmaceutics-17-00783-f004:**
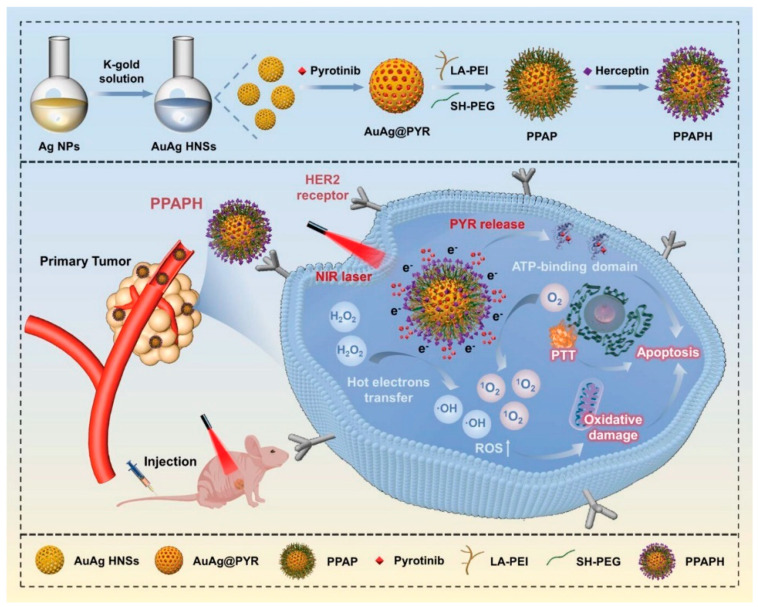
Procedure of PEG-PEI-AuAg@PYR@HCT (PPAPH) synthesis and PPAPH therapy for HER2 positive breast cancer. Hollow gold–silver nanocarriers with a porous structure for efficient Pyrotinib (PYR) loading were prepared using an electrodisplacement method. Premature drug release was prevented by surface modification with lipoic acid and polyethyleneimine (LA-PEI) and thiolated polyethylene glycol (SH-PEG). Subsequent modification with Herceptin (HCT) allowed precise targeting of HER2-overexpressing tumor cells. Localized, laser-induced hyperthermia and chemotherapy enhanced cytotoxicity and was highly effective in suppressing HER2-overexpressing BT474 cells. Tumor cell death was significantly promoted by the increase in ROS induced by AuAg hollow nanoshells. The synergistic effect of the nanosystem led to significant tumor shrinking in a mouse model of HER2-positive breast cancer. Reprinted with permission from Ref. [145]. Copyright 2024, John Wiley and Sons.

**Figure 5 pharmaceutics-17-00783-f005:**
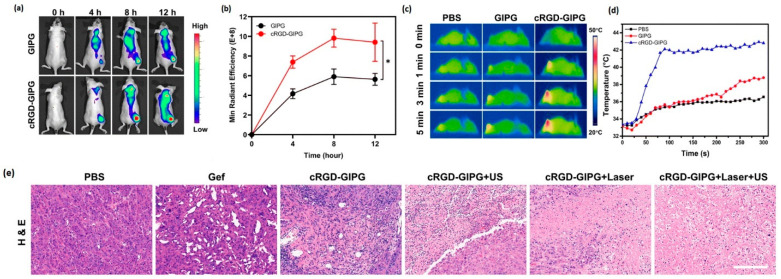
In vivo therapeutic effect of cRGD-GIPG NPs on a PC-9GR tumor-bearing mouse model. Real-time fluorescence images (**a**) demonstrate that GIPg and cRGD-GIPG rapidly accumulated in the tumor area after intravenous administration due to increased permeability. The fluorescence intensity (**b**) and accumulation rate in tumor sites were higher in the cRGD-GIPG group. Real-time infrared thermography (**c**) and temperature change curve (**d**) showed that cRGD-GIPG nanoparticles could significantly increase the tumor temperature to 42.8 °C after 5 min of irradiation, indicating nanoparticle accumulation in the tumor tissue and undergoing photothermal transformation. The H&E (**e**) staining assay was used to verify the therapeutic efficacy of different treatment groups demonstrated the most severe degree of apoptosis and cell necrosis in cRGD-GIPG+laser+US tissues. The reason behind this was that temperature change in the tumor area caused by laser irradiation was not uniform enough, while US irradiation reached deeper tissue layers and thus achieved a synergistic effect of the treatment (scale bar = 200 μm). Reprinted with permission from Ref. [148]. Copyright 2023, Royal Society of Chemistry.

**Figure 6 pharmaceutics-17-00783-f006:**
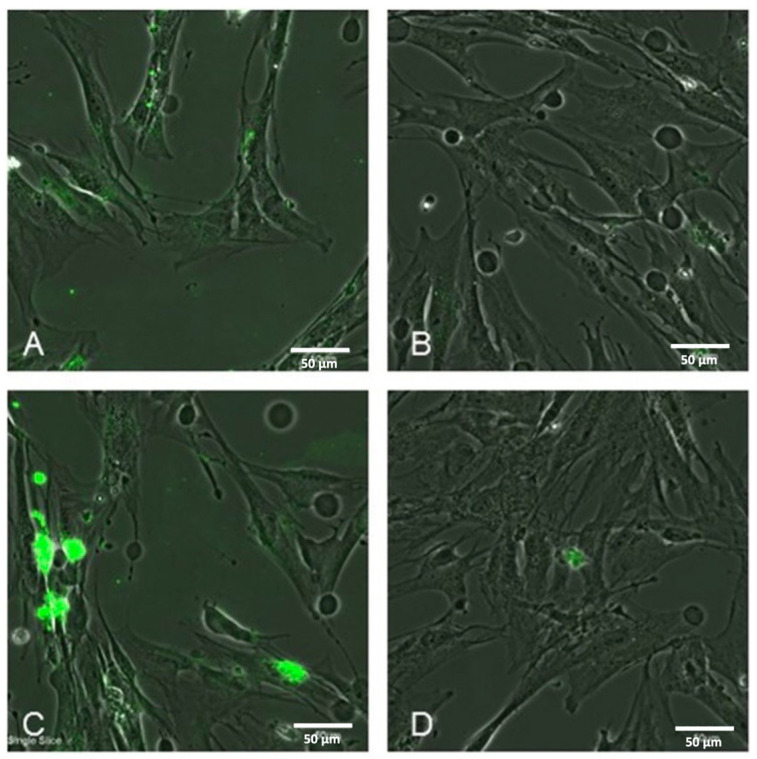
Confocal images of gold nanoparticles cell internalization. CD44 receptor-mediated nanocarrier internalization in pathological cells was demonstrated by treating lung fibroblasts with GNP-HC and GNP-IgG labeled with Alexa 488 fluorescent dye. Confocal microscopy demonstrated rapid selective internalization of GNP-HC in LF as evidenced by the green, fluorescent signal (**A**). GNPs-IgG were not internalized by the cells within 2 h of incubation (**B**) and the reason for this is that internalization requires the presence of a specific anti-CD44 antibody on the surface of the GNPs. Competition with specific antibody for CD44 formed an aggregate within the cells (**C**), and pretreatment with anti-CD44 did not alter the GNP-IgG-cell interaction (**D**). Reprinted with permission from Ref. [155]. Copyright 2019, Elsevier.

**Figure 7 pharmaceutics-17-00783-f007:**
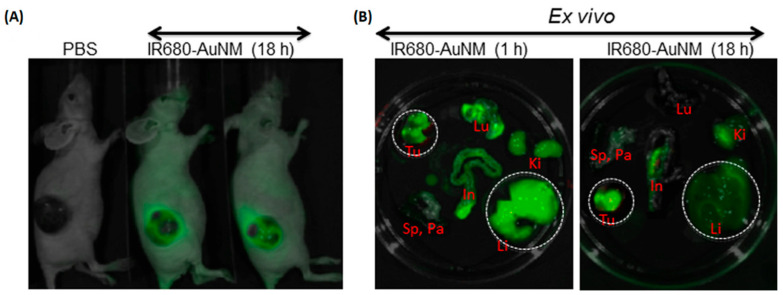
In vivo tumor delivery and biodistribution. Gold nanoparticles synthesized in micellar networks of amphiphilic block copolymer (AuNM) and conjugated with IR680 dye were fluorescently imaged (**A**) using IVIS spectra, while female mice with MDA-MB-231 human breast xenografts were injected with PBS or IR680-AuNM. After 18 h, a distinct fluorescent signal was observed in the tumor area. To further define AuNM biodistribution (**B**), ex vivo organ imaging was performed and the presence of AuNM was confirmed mainly in the liver (Li), lung (Lu), kidney (Ki) and intestine (In). Increased fluorescence signal was also observed in the tumor area, indicating the retention of Ir680-AuNM in the tissue. Reprinted with permission from Ref. [158]. Copyright 2017, American Chemical Society.

**Table 1 pharmaceutics-17-00783-t001:** Main characteristics of receptor tyrosine kinase families.

Type of RTKs	Name of the Receptor Family	Receptors	Function and Main Characterization	Reference
I	EPIDERMAL GROWTH FACTOR ErbB	ErbB-1 (HER1) or EGFR (*epidermal growth factor receptor)* ErbB-2 (HER2) ErbB-3 (HER3) ErbB-4 (HER4)	Regulation of cell growth, proliferation and migration of tumors. EGFR and HER2 are overexpressed in gastric cancer, HER3 preferentially activates the phosphatidylinositol 3-kinase (PI3K) pathway.	[14]
II	INSULIN RECEPTOR IR	IGF1R *(insulin-like growth factor I receptor)* InsR *(insulin receptor)* IRR *(IR-related receptor)*	Regulation of metabolism (main targets of action are liver, muscle and adipose tissue), growth, and proliferation. IR in the brain regulates cognitive behavior, food intake, dysfunction leads to diabetes, cancer or Alzheimer’s disease.	[15]
III	PDGFR, CSFR, Kit, FLT3	PDGFR α/β *(platelet-derived growth factor receptor α/β)* CSF1R *(colony-stimulating factor 1 receptor)* c-Kit FLT3 *(fms-related tyrosine kinase 3)*	Mutations have a major impact on leukemic transformation of acute myeloid leukemia (AML) cells. PDGFR α/β also regulates bone formation, tissue repair, and fibroblast proliferation.	[16]
IV	VASCULAR ENDOTHELIAL GROWTH FACTOR VEGF	VEGFR-1 or Flt-1 VEGFR-2 or KDR VEGFR-3 or Flt-4 *(vascular endothelial growth factor receptors 1/2/3) or (fms related receptor tyrosine kinase 1/4 and kinase insert domain receptor)*	Regulation of tumor-induced angiogenesis. VEGFRs are essential for the development of hematopoietic cells, vascular endothelial cells and lymphatic endothelial cells, VEGFR3 plays a critical role in lymphangiogenesis and the spread of tumor cells to regional lymph nodes.	[17]
V	FIBROBLAST GROWTH FACTOR FGF	FGFR1/2/3/4 *(fibroblast growth factor receptor 1/2/3/4)*	Promotion of cell survival, proliferation, development, angiogenesis and differentiation. Highest alteration frequency of FGFR was found in urothelial cancer, cholangiocarcinoma, endometrial cancer, squamous lung cancers, breast cancer and cervical cancer.	[18]
VI	PTK7/CCK4	PTK7 *(tyrosine-protein kinase-like 7)* or CCK4 *(colon carcinoma kinase 4)*	PTK7 influences the establishment of cell polarity, regulation of cell movement and migration and cell invasion. Pseudotyrosine kinase PTK7 is overexpressed in several solid tumors and hematological malignancies and linked to metastasis, poor prognosis, and resistance to treatment.	[19]
VII	TROPOMYOSIN RECEPTOR KINASE TRK	TRKA/B/C	TRKs are encoded by the NTRK genes (neurotrophins) and play a role in the development and normal functioning of the nervous system. NTRK gene fusions occur in thyroid cancer, colorectal and appendiceal cancer, lung cancer, sarcoma, central nervous system or gastrointestinal stromal tumors.	[20]
VIII	ROR	ROR1/2 *(receptor tyrosine kinase like orphan receptor 1/2)*	Regulation of cell polarity, migration, proliferation and differentiation during developmental morphogenesis, tissue-/organo-genesis and regeneration of adult tissues following injury. RORs are implicated in age-related diseases, including tissue fibrosis, atherosclerosis (or arteriosclerosis), neurodegenerative diseases, and cancers.	[21]
IX	MuSK	MuSK *(muscle-specific tyrosine kinase receptor)*	Regulation of formation and stabilization of neuromuscular junctions (NMJs). MuSK is expressed in mammalian tissues other than skeletal muscle, including excitatory neurons in the central nervous system.	[22]
X	HEPATOCYTE GROWTH FACTOR HGF or SCATTER FACTOR SF	MET *(proto-oncogene tyrosine kinase receptor)*	MET is expressed in all human cell types, overexpressed in multisystem tumors, including respiratory, digestive, reproductive, nervous and epithelial tissue tumors.	[23]
XI	TAM	AXL TYRO3 MER	Alteration of TAM receptor function can lead to autoimmune disease, retinitis pigmentosa, and cancers (myeloid and lymphoblastic leukemias, melanoma, breast, lung, colon, liver, gastric, kidney, ovarian, uterine, and brain)	[24]
XII	TIE	TIE1/2	TIEs are expressed in endothelial cells and are key regulators of normal blood and lymphatic vessel development and of pathological processes, including tumor angiogenesis (Lewis lung carcinoma, melanoma, EL4 leukemia/lymphoma), progression and metastasis, atherosclerosis, and vascular leakage.	[25]
XIII	EPH RECEPTOR- -INTERACTING PROTEIN Ephrin	EphA1 to EphA8 and inactive EphA10 EphB1 to EphB4 and inactive EphB6 *(erythroprotein-* *-producing human hepatocellular receptors)*	The largest of the RTK families. Ephs are expressed in most adult tissues and on immune system cells and have complex roles in embryonic and neural developmental processes such as cell segregation and migration, spatial organization of cell populations, tissue boundary formation, axonal guidance, and angiogenesis. Eph receptors are involved in the pathogenesis of various diseases, e.g., atherosclerosis, fibrosis, CNS diseases and cancer (EphA10 expressed on breast cancer cells)	[26]
XIV	RET	RET *(proto-oncogene receptor)*	Mutations in the RET gene have been found in several different cancers of neuroendocrine origin (papillary thyroid carcinoma, medullary thyroid carcinoma, multiple endocrine neoplasias) and a gut syndrome characterized by intestinal obstruction known as Hirschsprung’s disease.	[27]
XV	RYK	RYK *(receptor like tyrosine kinase)*	RYK is highly expressed in various malignancies, including mesothelioma, small cell lung cancer, gastric cancer, glioblastoma, liver cancer, acute leukemias and breast cancer.	[28]
XVI	DDR	DDR1/2 *(discoidin domain receptor tyrosine kinase 1/2)*	DDRs recognize collagens as their ligands and regulate cell-collagen interactions in normal and pathological conditions.	[29]
XVII	ROS	ROS1 *(proto-oncogene 1)*	ROS1 overexpression is observed in 80−100% of metastatic oral squamous cell carcinomas, upregulated in E-cadherin-deficient breast cancers, glioblastoma, NSCLC, Spitzoid neoplasms and inflammatory myofibroblastic tumors.	[30]
XVIII	LMK	LMTK1-a LMTK1-b LMTK2 LMTK3 *(lemur tyrosine kinase 1-a/1-b/2/3)*	Regulation of axonal transport and endosomal trafficking, modulation of synaptic functions, memory and learning. LMTKs are involved in various diseases including cancer (breast, prostate, lung, colorectal, renal, testis and ovarian, thyroid, pancreatic, bladder, gastric, glio- and neuroblastoma, and leukemia), cystic fibrosis, Alzheimer’s disease, amyotrophic lateral sclerosis/frontotemporal dementia and global developmental delay/intellectual disability.	[31]
XIX	LTK	LTK *(leukocyte receptor tyrosine kinase)* ALK *(anaplastic lymphoma kinase receptor)*	Very little is known about the physiological role of LTK tyrosine kinase. The LTK gene is preferentially expressed in leukemias with no cell lineage specificity, but not in other neoplasms. Most mutations of the ALK gene are in the form of a translocation with another partner gene leading to a fusion oncogene. ALK-rearrangement was identified in many different cancers, including inflammatory myofibroblastic tumors, diffuse large B-cell lymphoma, non-small-cell lung cancer, and esophageal squamous cell, colorectal, and breast carcinomas.	[32,33]
XX	STYK	STYK1 *(serine/threonine/* *tyrosine kinase 1)*	STYK1 is a potent oncogene that enhances cell proliferation in vitro and drives both tumorigenesis and metastasis in animal model systems and aberrant expression has been identified in a wide range of cancer types, including lung, ovarian, breast, colorectal, prostate and renal cell cancer.	[34]

**Table 2 pharmaceutics-17-00783-t002:** Approved receptor tyrosine kinase inhibitors, their main targets and indications (information obtained from the official FDA [51] and EMA [52] websites on 5 Oct 2024).

	International Name	Brand Name	Company	Year Approved		Primary Targets	Therapeutic Indications
				FDA	EMA		
1	Fruquintinib	Fruzaqla	Takeda Pharma	2023	2024	VEGFR	Metastatic colorectal cancer
2	Quizartinib	Vanflyta	Daiichi Sankyo	2023	2023	FLT3	Acute myeloid leukemia
3	Repotrectinib	Augtyro	Bristol Myers	2023	disapproved	ROS1	Non-small-cell lung carcinoma
4	Futibatinib	Lytgobi	Taiho Pharma Netherlands B.V.	2022	2023	FGFR2	Cholangiocarcinoma
5	Infigratinib	Truseltiq	QED Therapeutics	2021 ^a^	-	FGFR2	Cholangiocarcinoma
6	Mobocertinib	Exkivity	Takeda Pharma	2021	application withdrawn	EGFR	Non-small-cell lung carcinoma
7	Tepotinib	Tepmetko	Merck	2021	2022	MET	Non-small-cell lung carcinoma
8	Tivozanib	Fotivda	AVEO Pharma	2021	2017	VEGFR	Kidney cancer
9	Avapritinib	Ayvakyt	Blueprint Medicines	2020	2020	PDGFRα KIT	Gastrointestinal stromal tumors Mastocytosis
10	Capmatinib	Tabrecta	Novartis	2020	2022	MET	Non-small-cell lung carcinoma
11	Pemigatinib	Pemazyre	Incyte	2020	2021	FGFR	Cholangiocarcinoma
12	Pralsetinib	Gavreto	Rigel Pharma	2020	2021	RET	Non-small-cell lung carcinoma
13	Ripretinib	Qinlock	Decipera Pharma	2020	2021	KIT PDGFRα	Gastrointestinal stromal tumor Stomach and bowel cancer
14	Selpercatinib	Retsevmo	Eli Lilly	2020	2021	RET	Non-small-cell lung carcinoma Thyroid cancer Solid tumors
15	Tucatinib	Tukysa	Pfizer	2020	2021	ErbB2/HER2	HER2-positive breast cancer
16	Entrectinib	Rozlytrek	Genentech (Roche)	2019	2020	TRK ROS1	Solid tumors Non-small-cell lung carcinoma
17	Erdafitinib	Balversa	Janssen Pharma	2019	2024	FGFR	Urothelial bladder and urinary cancer
18	Pexidartinib	Turalio	Daiichi Sankyo	2019	refused	CSF1R KIT FLT3	Tenosynovial giant cell tumors
19	Ftinib	Vizimpro	Pfizer	2018	2019	EGFR	Non-small-cell lung carcinoma
20	Gilteritinib	Xospata	Astellas Pharma	2018	2019	FLT3	Acute myeloid leukemia
21	Larotrectinib	Vitrakvi	Bayer	2018	2019	TRK	Lungs, thyroid glands and intestines carcinomas
22	Lorlatinib	Lorbrena (US, Canada, Japan) Lorviqua (EU)	Pfizer	2018	2019	ALK	Non-small-cell lung carcinoma
23	Brigatinib	Alunbrig	Takeda Pharma	2017	2018	ALK	Non-small-cell lung carcinoma
24	Midostaurin	Rydapt	Novartis	2017	2017	FLT3 KIT	Acute myeloid leukemia
25	Neratinib	Nerlynx	Puma Biotech	2017	2018	ErbB2/HER2	HER2-positive breast cancer
26	Alectinib	Alecensa	Roche	2015	2017	ALK	Non-small-cell lung carcinoma
27	Lenvatinib	Lenvima	Easai	2015	2015	VEGFR FGFR RET	Thyroid Neoplasms Hepatocellular carcinoma Endometrial carcinoma
28	Osimertinib	Tagrisso	AstraZeneca	2015	2016	EGFR	Non-small-cell lung carcinoma
29	Ceritinib	Zykadia	Novartis	2014	2015	ALK	Non-small-cell lung carcinoma
30	Nintedanib	Vargatef	Boehringer Ingelheim	2014	2014	VEGFR FGFR PDGFR	Non-small-cell lung carcinoma
31	Afatinib	Giotrif	Boehringer Ingelheim	2013	2013	ErbB	Non-small-cell lung carcinoma
32	Axitinib	Inlyta	Pfizer	2012	2012	VEGFR	Renal cell carcinoma
33	Cabozantinib	Cometriq capsule Cabometyx tablet form	Exelixis	2012	2014	VEGFR MET RET	Thyroid neoplasms Renal cell carcinoma Hepatocellular carcinoma
34	Regorafenib	Stivarga	Bayer	2012	2013	VEGFR	Colorectal cancer Gastrointestinal stromal tumor Hepatocellular carcinoma
35	Crizotinib	Xalkori	Pfizer	2011	2012	ALK ROS1	Non-small-cell lung carcinoma
36	Vandetanib	Caprelsa (US) Zactima (EU)	Sanofi	2011	application withdrawn	EGFR VEGFR	Medullary thyroid cancer
37	Pazopanib	Votrient	GSK	2009	2010	VEGFR PDGFR FGFR	Renal cell carcinoma Soft-tissue sarcomas
38	Lapatinib	Tyverb	GSK	2007	2008	ErbB2/HER2	Breast neoplasms
39	Sunitinib	Sutent	Pfizer	2006	2006	PDGFR VEGFR KIT	Gastrointestinal stromal tumors Renal cell carcinoma Neuroendocrine tumors
40	Sorafenib	Nexavar	Bayer	2005	2006	VEGFR KIT FLT3	Hepatocellular carcinoma Renal cell carcinoma Differentiated thyroid carcinoma
41	Erlotinib	Tarceva	Genentech (Roche Group)	2004	2005	EGFR	Non-small-cell lung carcinoma Pancreatic neoplasms
42	Gefitinib	Iressa	AstraZeneca	2003	2009	EGFR	Non-small-cell lung carcinoma

^a^ 16 May 2024 FDA final withdrawal decision [53].

**Table 3 pharmaceutics-17-00783-t003:** Approved non-receptor tyrosine kinase inhibitors, their main targets and indications (information obtained from the official FDA [51] and EMA [52] websites on 5 Oct 2024).

	International Name	Name of Medicine	Company	Year Approved		Primary Targets	Therapeutic Indications
				FDA	EMA		
1	Momelotinib	Ojjaara (US) Omjjara (EU)	GlaxoSmith Kline	2023	**2024**	JAK	Splenomegaly Myeloproliferative disorders
2	Pirtobrutinib	Jaypirca	Eli Lilly	2023	2023	BTK	Mantle cell lymphoma
3	Ritlecitinib	Litfulo	Pfizer	2023	2023	JAK3	Alopecia areata
4	Abrocitinib	Cibinqo	Pfizer	2022	2021	JAK	Atopic dermatitis
5	Deucravacitinib	Sotyktu	Bristol-Myers Squibb	2022	2023	TYK2 JAK	Psoriasis
6	Pacritinib	Vonjo	CTI Biopharma	2022	disapproved	JAK	Post-polycythemia vera Post-essential thrombocythemia
7	Asciminib	Scemblix	Novartis	2021	2022	BCR-ABL	Chronic myeloid leukemia
8	Fedratinib	Inrebic	Bristol-Myers Squibb	2019	2021	JAK	Myelofibrosis
9	Zanubrutinib	Brukinsa	BeiGene USA	2019	2021	BTK	Waldenström’s macroglobulinemia Marginal zone lymphoma Chronic lymphocytic leukemia Follicular lymphoma
10	Upadacitinib	Rinvoq	AbbVie	2019	2019	JAK	Rheumatoid arthritis Psoriatic arthritis Atopic dermatitis Axial spondyloarthritis Ulcerative colitis
11	Fostamatinib	Tavalisse (US) Tavlesse (EU)	Rigel Pharma	2018	2020	SYK (SRC family)	Chronic immune thrombocytopenia
12	Baricitinib	Olumiant	Eli Lilly	2018	2017	JAK	Rheumatoid arthritis Atopic dermatitis Alopecia areata Juvenile idiopathic arthritis
13	Acalabrutinib	Calquence	Astra Zeneca	2017	2020	BTK	Chronic lymphocytic leukemia Blood cancer affecting B cells
14	Ibrutinib	Imbruvica	Pharmacyclics LLC	2013	2014	BTK	Mantle cell lymphoma Chronic lymphocytic leukemia Waldenström’s macroglobulinemia
15	Bosutinib	Bosulif	Pfizer	2012	2013	BCR-ABL	Chronic lymphocytic leukemia
16	Ponatinib	Iclusig	Takeda Pharms	2012	2013	BCR-ABL	Chronic lymphocytic leukemia Acute lymphoblastic leukemia
17	Tofacitinib	Xeljanz	Pfizer	2012	2017	JAK	Rheumatoid arthritis Psoriatic arthritis Juvenile idiopathic arthritis Ulcerative colitis Ankylosing spondylitis
18	Ruxolitinib	Jakafi (US) Jakavi (EU)	Incyte	2011	2012	JAK	Splenomegaly Polycythemia vera Acute or chronic graft-versus-host disease
19	Nilotinib	Tasigna	Novartis	2007	2007	BCR-ABL	Chronic lymphocytic leukemia
20	Dasatinib	Sprycel	Bristol Myers Squibb	2006	2006	BCR-ABL	Chronic myeloid leukemia Ph+ acute lymphoblastic leukemia
21	Imatinib	Gleevec (US) Glivec (EU)	Novartis	2001	2001	BCR-ABL	Chronic myeloid leukemia Ph+ acute lymphoblastic leukemia Myeloproliferative diseases Advanced hypereosinophilic syndrome Gastrointestinal stromal tumors Dermatofibrosarcoma protuberans

**Table 4 pharmaceutics-17-00783-t004:** Conjugates of metal nanoparticles with TKIs, their use in anticancer therapy and enhancements of the nanodrug delivery system compared to the TKI itself.

Type of Metal NPs	TKI	Treatment	Nanosystem Enhancement Effect	Reference, Year
platinum	sunitinib	Renal fibrosis	Accelerated drug accumulation, longer time in circulation	[131], 2012
silver	erlotinib	- *(SERS analysis)*	-	[135], 2018
	imatinib	- *(X-ray CT imaging,* *FCS measurement)*	-	[137], 2017
gold	midostaurin	Acute myeloid leukemia (AML)	Inhibition of tumor formation, increased cytotoxicity against tumor cells	[144], 2024
	pyrotinib	Early breast cancer (EBC)	Inhibition of cell proliferation, photothermal effect, tumor shrinkage Inhibition of cell proliferation, photothermal effect, tumor shrinkage Inhibition of cell proliferation, photothermal effect, tumor shrinkage Inhibition of cell proliferation, photothermal effect, tumor shrinkage	[145], 2024
	gefitinib	Non-small cell lung cancer (NSCLC)	Almost complete removal of tumor tissue by the photothermal effect, reducing the viability of the tumor cell line	[148], 2023
	sorafenib	Renal cancer	High percentage of drug released, tumor growth suppression, reduction in hepatotoxicity High percentage of drug released, tumor growth suppression, reduction of hepatotoxicity	[149], 2021
	icotinib	- *(UV/vis spectroscopy, TEM,* *DFT, SERS analysis)* *(SERS analysis)*	-	[165], 2021
	SI306	Glioblastoma (GBM)	Reduction in tumor cell viability in combination with radiotherapy	[150], 2020
	varlitinib	Pancreatic cancer	Reduction in tumor cell viability	[152], 2019
	imatinib	Scleroderma-associated interstitial lung disease (SSc-ILD)	Controlled drug release, selective drug activity against c-Abl and PDGFR	[155], 2019
	afatinib	Non-small cell lung cancer (NSCLC)	Improved biocompatibility and efficacy of the drug, reduced tumor cell viability	[156], 2019
	nilotinib	- *(Secondary ion mass* *spectrometry imaging)*	-	[157], 2017
	vandetanib	Metastatic breast cancer (MBC)	Reduction in migration and viability of tumor cells, induction of apoptosis	[158], 2017
	imatinib	Chronic myelogenous leukemia (CML)	Decrease in IC50 of the drug, decrease in viability of drug-resistant cells	[159], 2017
	midostaurin sorafenib lestaurtinib quizartinib	Acute myeloid leukemia (AML)	Decrease in the viability of tumor cells due to the effect of the gelatinous coating	[161], 2016
	dasatinib	Acute myeloid leukemia (AML)	Increase in drug efficacy, decrease in toxicity	[162], 2016
	sorafenib	Renal cell carcinoma (RCC)	Necrosis induced by the photothermal effect, significant reduction in the tumor	[163], 2016

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
