# Peer review of "Innovative Approaches in Cancer Treatment: Emphasizing the Role of Nanomaterials in Tyrosine Kinase Inhibition"

_pharmaceutics, 2025, doi:10.3390/pharmaceutics17060783_

Round 1
Reviewer 1 Report
Comments and Suggestions for Authors
The article is a timely and contemplative summary of the role of nanomaterials in enhancing the inhibition of tyrosine kinases as a therapeutic approach for cancer. The topic is particularly pertinent given the present challenges of drug resistance and toxicity of conventional tyrosine kinase inhibitors. However, some aspects of the review require additional clarification, description, or detail to further enhance the scientific content and overall effect of the review.
- A 27% similarity index (a plagiarism measure) is higher than what is acceptable for publication and therefore raises concerns about originality and requires a great deal of revisions in order to satisfy ethical standards. It is recommended that the authors significantly revise the manuscript to reduce the similarity index to 15–20%, thereby maintaining originality while appropriately citing the utilized sources.
- The manuscript does not have an explicitly described study selection methodology and analysis of studies covered in this review. While it is a narrative review, it would be better if there is a concise description of the literature search strategy (databases used, keywords, time period, inclusion and exclusion criteria). Add a section or paragraph describing how the literature was selected and analyzed to make it more transparent and reproducible.
- Please differentiate from previous reviews and highlighting new developments. Please clearly state in the introduction the distinctions between this paper and earlier reviews, and indicate the novelty and original contributions of this paper.
- Indicate whether the review includes both small-molecule tyrosine kinases inhibitors and monoclonal antibodies, or if it is restricted to small-molecule inhibitors.
- Explain the nanomaterials employed for the delivery of tyrosine kinases inhibitors and their physicochemical properties controlling biodistribution, targeting, and safety.
- Provide at least one comprehensive table or schematic comparing tyrosine kinases inhibitors alone versus tyrosine kinases inhibitors-loaded nanocarriers in terms of delivery efficiency, toxicity, and tumor response in preclinical/clinical trials.
- Significant contributions have been made in the article; however, in order to enhance the discussions and give a more tailored context, it is better for the authors to reconsider and cite the following relevant studies in relation to cancer drug delivery- “Rational combinations of targeted cancer therapies: background, advances and challenges.”, “Nanocarrier cancer therapeutics with functional stimuli-responsive mechanisms.”, “Development, optimization, and characterization of polymeric micelles to improve dasatinib oral bioavailability: Hep G2 cell cytotoxicity and in vivo pharmacokinetics for targeted liver cancer therapy.”, “Smart Nanoplatforms Responding to the Tumor Microenvironment for Precise Drug Delivery in Cancer Therapy.” Nanocarriers for cancer nano-immunotherapy.’’, “Tumor Microenvironment-Based Stimuli-Responsive Nanoparticles for Controlled Release of Drugs in Cancer Therapy.”, “Nanostructured Cubosomes in a Thermoresponsive Depot System: An Alternative Approach for the Controlled Delivery of Docetaxel.”, Harnessing the Power of Stimuli-Responsive Nanoparticles as an Effective Therapeutic Drug Delivery System.” “Cabozantinib-phospholipid complex for enhanced solubility, bioavailability, and reduced toxicity in liver cancer.”, “Nanoparticle-Based Drug Delivery Systems Targeting Tumor Microenvironment for Cancer Immunotherapy Resistance: Current Advances and Applications.”
- Although resistance is briefly mentioned, the article does not touch upon the molecular determinants of resistance to tyrosine kinases inhibitors (e.g., gatekeeper mutations, bypass signaling, efflux transporters), much less how nanomedicine targets them specifically.
- A special section on how nanomaterials can bypass resistance mechanisms (e.g., efflux pumps, kinase mutations) would add to the review.
- While the preclinical efficacy is highlighted, the clinical development of nanomaterial-tyrosine kinases inhibitors treatments needs to be addressed
- The conclusion is ambiguous and does not summarize the subtle challenges (e.g., regulatory hurdles, scale-up, immunogenicity) to clinical translation of nanomedicine.
- Ensure all tables and figures are legible and caption them appropriately with suitable detailed captions. All figures should be mentioned in the text and to be addressed.
- English must also be enhanced to obtain clarity and to avoid typos and grammatical mistakes.
The paper is a milestone work in oncology and nanomedicine, and it targets a significant therapeutic issue. In response to the above comments, namely by refining the completeness of the results section, elucidating the methodology, and adding visual supports, the paper will be significantly improved.
Comments on the Quality of English LanguageEnglish must also be enhanced to obtain clarity and to avoid typos and grammatical mistakes.
Reviewer 2 Report
Comments and Suggestions for Authors
This s a thoroughly researched and well-written review that covers an important topic of nanoparticle (NP) delivery systems for tyrosine kinase inhibitors (TKIs), which are currently the leading class of anti-cancer drugs. Sections 1-2 mostly give a general overview of tyrosine kinases and their inhibitors, and Section 3 gives an overview of the NP delivery systems, before the two topics are combined in Section 4. The introductory part (Sections 1-3) may seem too long, but its length is justified by clear explanation of the existing problems in TKIs and a need for the development of better delivery systems, which will be particularly useful for non-specialists in the field. Section 4 provides a comprehensive review of the currently developed TKI-NP systems, complemented with a useful summary table and illustrations taken from the primary publications. My only concern is the partial overlap between Sections 3 and 4, which is reflected in their similar titles. Section 4 should be limited to TKI-NP systems, while Section 3 should provide a general introduction into NP as drug delivery systems. In addition, permission should be sought for reproducing any images from primary publications, even if they were modified.
Round 2
Reviewer 1 Report
Comments and Suggestions for Authors
I have thoroughly reviewed the revised version of the manuscript titled "Innovative Approaches in Cancer Treatment: Emphasizing the Role of Nanomaterials in Tyrosine Kinase Inhibition." The authors have satisfactorily addressed all the comments and concerns raised during the review process.
To further enhance the scientific rigor and relevance of this review article, I recommend referencing and citing a recently published review titled "Natural Macromolecules Polysaccharide-Based Drug Delivery Systems Targeting Tumor Necrosis Factor Alpha Receptor for the Treatment of Cancer: A Review." (https://doi.org/10.1016/j.ijbiomac.2025.145145) This work provides valuable insights into macromolecule-based drug delivery systems targeting TNF-α in cancer therapy, which aligns well with the scope of the current manuscript. Its inclusion would strengthen the discussion on advanced therapeutic strategies in cancer treatment.
